# A haemagglutination test for rapid detection of antibodies to SARS-CoV-2

Alain Townsend [1,2✉], Pramila Rijal[1,2], Julie Xiao[1], Tiong Kit Tan [1], Kuan-Ying A. Huang [3,4], Lisa Schimanski[1,2], Jiandong Huo[5], Nimesh Gupta [6], Rolle Rahikainen [7], Philippa C. Matthews[8,9], Derrick Crook[8,10,11], Sarah Hoosdally[9,11], Susanna Dunachie [9,12], Eleanor Barnes [10], Teresa Street[10,11], Christopher P. Conlon [10], John Frater [10], Carolina V. Arancibia-Cárcamo[10], Justine Rudkin[13], Nicole Stoesser [8,10], Fredrik Karpe[11,14], Matthew Neville [14], Rutger Ploeg [15], Marta Oliveira[15], David J. Roberts[16,17], Abigail A. Lamikanra[16], Hoi Pat Tsang[16], Abbie Bown[18], Richard Vipond[18], Alexander J. Mentzer [19], Julian C. Knight [2,11,19], Andrew J. Kwok [19], Gavin R. Screaton [19,20], Juthathip Mongkolsapaya [2,19,21], Wanwisa Dejnirattisai[19], Piyada Supasa[19], Paul Klenerman[9], Christina Dold[22,23], J. Kenneth Baillie [24], Shona C. Moore [25], Peter J. M. Openshaw [26], Malcolm G. Semple [25], Lance C. W. Turtle [25], Mark Ainsworth[12], Alice Allcock[19], Sally Beer [12], Sagida Bibi[22], Donal Skelly [12], Lizzy Stafford[12], Katie Jeffrey [12], Denise O'Donnell[12], Elizabeth Clutterbuck[22], Alexis Espinosa[12], Maria Mendoza[12], Dominique Georgiou[12], Teresa Lockett[12], Jose Martinez[12], Elena Perez[12], Veronica Gallardo Sanchez[12], Giuseppe Scozzafava[12], Alberto Sobrinodiaz[12], Hannah Thraves[12] & Etienne Joly [27✉]

Serological detection of antibodies to SARS-CoV-2 is essential for establishing rates of seroconversion in populations, and for seeking evidence for a level of antibody that may be protective against COVID-19 disease. Several high-performance commercial tests have been described, but these require centralised laboratory facilities that are comparatively expensive, and therefore not available universally. Red cell agglutination tests do not require special equipment, are read by eye, have short development times, low cost and can be applied at the Point of Care. Here we describe a quantitative Haemagglutination test (HAT) for the detection of antibodies to the receptor binding domain of the SARS-CoV-2 spike protein. The HAT has a sensitivity of 90% and specificity of 99% for detection of antibodies after a PCR diagnosed infection. We will supply aliquots of the test reagent sufficient for ten thousand test wells free of charge to qualified research groups anywhere in the world.

A full list of author affiliations appears at the end of the paper.

Red cell agglutination tests have a distinguished history. Since Landsteiner's classic observations in 1901[1] (English translation), they have been used for the determination of blood groups[2], detection of influenza viruses,[3] and in a wide variety of applications championed by Prof. Robin Coombs for the detection of specific antibodies or antigens[4] (reviewed by ref. [5]). They have the great advantage of being simple, inexpensive, can be read by eye, and do not require sophisticated technology for their application. In the recent era, the linkage of an antigen to the red cell surface has become easier with the possibility of fusing a protein antigen sequence with that of a single domain antibody or nanobody specific for a molecule on the red cell surface (discussed in ref. [6]).

We have applied this concept to provide a simple haemagglutination test ("HAT") for the detection of antibodies to the receptor binding domain (RBD) of the SARS-CoV-2 spike protein. The RBD is a motile subdomain at the tip of the SARS-CoV-2 spike protein that is responsible for binding the virus to its ACE2 receptor. The RBD of betacoronaviruses folds independently of the rest of the spike protein[7–10]. This useful property provides an Achilles' heel for the virus and allows many potential applications in vaccine design[11–16] and serology[17–19], see also www.gov.uk/government/publications/COVID-19-laboratory-evaluations-of-serological-assays. The majority of neutralising antibodies bind to the RBD[18,20], and the level of antibody to the RBD detected in ELISA correlates with that of neutralising antibodies[17,18,21,22]. We reasoned therefore that a widely applicable and inexpensive test for antibodies to the RBD would be useful for research in settings where high throughput assays were not available.

In order to link the SARS-CoV-2 RBD to red cells, we selected the single domain antibody (nanobody) IH4[6], specific for a conserved epitope on glycophorin A. Glycophorin A is expressed at up to $10^6$ copies per red cell. The IH4 nanobody has previously been linked to HIV p24 to provide a monomeric reagent that bound p24 to the red cell surface. Antibodies to p24 present in serum crosslinked the p24 and agglutinated the red cells[6]. We have adapted this approach for detection of antibodies to SARS-CoV-2 by linking the RBD of the SARS-CoV-2 spike protein to IH4 via a short (GSG)2 linker to produce the fusion protein IH4-RBD-6H (Fig. 1). Since we embarked on this project, another group has described preliminary results with an approach similar to ours, but using a fusion of the RBD to an scFv against the H antigen to coat red blood cells with the SARS-2 RBD[23].

## Results

**Production of the IH4-RBD Reagent**. The IH4-RBD sequence (Fig. 1B and Supplementary Note) was codon optimised and expressed in Expi293F cells in a standard expression vector (available on request). One advantage of this mode of production compared to bacterially produced protein as used by Habib et al. is that the reagent will carry the glycosylation moieties found in humans, which may play a role in the antigenicity of the RBD[24]. The protein (with a 6xHis tag at the C-terminus for purification) was purified by Ni-NTA chromatography which yielded ~160 mg/L. We later had 1 g of the protein synthesised commercially by Absolute Antibody, Oxford. The IH4-RBD protein ran as single band at ~40 kDa on SDS-PAGE (Fig. 1C).

**Establishment of the HAT with monoclonal antibodies to the RBD**. One purpose envisaged for the HAT is for use as an inexpensive point of care test for detection of antibodies in capillary blood samples obtained by a "finger prick". We therefore wished to employ human red cells as indicators without the need for cell separation or washes, to mimic this setting. The use of V-bottom microtiter plates to perform simplified hemagglutination

tests was first described over 50 years ago[25]. In preliminary tests, we observed that 50 μL of whole blood (K2EDTA sample) diluted 1:40 in phosphate-buffered saline (PBS), placed in V-bottomed wells of a standard 96-well plate, settled in 1 h to form a button of red cells at the bottom of the well. The normal haematocrit of blood is ~40% vol/vol, so this dilution provides ~1% red cells. If the plate was then tilted, the red cell button flowed to form a "teardrop" in ~30 s (for example, Fig. 2A, bottom row).

If serum or plasma samples are to be tested, a standard collection of 10 mL of type O Rh-negative (O−ve) blood into a K2EDTA tube will thus provide sufficient red cells for 8000 test wells.

A well-characterised monoclonal antibody to the RBD, CR3022[26,27] added to the red cells at between 0.5 and 32 ng/well in 50 μL, did not agglutinate the cells on its own (Fig. 2A, row 8). The addition of the IH4-RBD reagent at between 12.5 and 800 ng/well (in 50 μL PBS) induced a concentration dependent agglutination of the red cells, detected by the formation of a visible mat or plug of agglutinated cells, and the loss of teardrop formation on tilting the plate (Fig. 2A). From repeated trials of this experiment, we established that a standard addition of 100 ng/well of the IH4-RBD developer (50 μL of a stock solution of 2 μg/mL in PBS) induced agglutination of 50 μL of 1:40 human red cells in the presence of as little as 2 ng/well of the CR3022 monoclonal antibody. The standardised protocol used for the subsequent tests were thus performed in 100 or 150 μL final volume, containing 100 ng of the IH4-RBD developer, and 50 μL 1:40 whole blood (~1% v/v red cells ~0.5 μL packed red cells per reaction). After 60 min incubation at room temperature, we routinely photographed the plates after the 30 s tilt for examination and reading.

The requirement for 100 ng of the IH4-RBD developer per test well means that the gram of IH4-RBD protein we have had synthesised is sufficient for 10 million test wells at a cost of ~0.27 UK pence per test.

Having established a standard addition of 100 ng/well of the IH4-RBD reagent, we screened a set of 13 human monoclonal antibodies, two divalent nanobodies, divalent ACE2-Fc, and a murine monoclonal antibody to the 6H tag, which are known to bind to the RBD[24,28–31]. These reagents bind to at least four independent sites on the RBD, and some are strongly neutralising and capable of profound ACE2 blockade[28,29]. Overall, 13 of the 15 divalent molecules agglutinated red cells and titrated in the HAT to an endpoint between 2 and 125 ng/well, after addition of 100 ng IH4-RBD (Fig. 2B and Supplementary Table 1). One monoclonal antibody, FD-5D [28] and one divalent nanobody, VHH72-Fc[31], failed to agglutinate red cells in the presence of the IH4-RBD reagent. However, if a monoclonal antibody to human IgG was added to the reaction (50 μL of mouse anti-human IgG, Sigma Clone GG-5 1:100), these molecules specifically agglutinated the red cells (Supplementary Fig. 1A, B). This result, analogous to the "indirect" Coombs test[4,5], suggested that these two molecules had bound to the RBD associated with the red cells but failed to crosslink to RBDs on neighbouring red cells. However, these could be crosslinked by the anti-IgG reagent. Monoclonal antibodies to other regions of the spike protein (EW-9B, EW-9C, and FJ-1C) failed to agglutinate red cells (Fig. 2B, Supplementary Table 1). Finally, we looked at the effect of a divalent ACE2-Fc molecule constructed by fusing the peptidase domain of ACE2 (amino acids 19–615) to the hinge and Fc region of human IgG1 (described in ref. [29]). ACE2-Fc agglutinated red cells strongly in the presence of 100 ng/well of the IH4-RBD developer, titrating to ~4 ng/well (Fig. 2B, rows 7 and 8).

In summary, these results showed that all of the known epitopes bound by characterised monoclonal antibodies were displayed by the IH4-RBD reagent, as well as the ACE2 binding site, and could mediate agglutination by specific antibodies, divalent nanobodies, or ACE2-Fc.

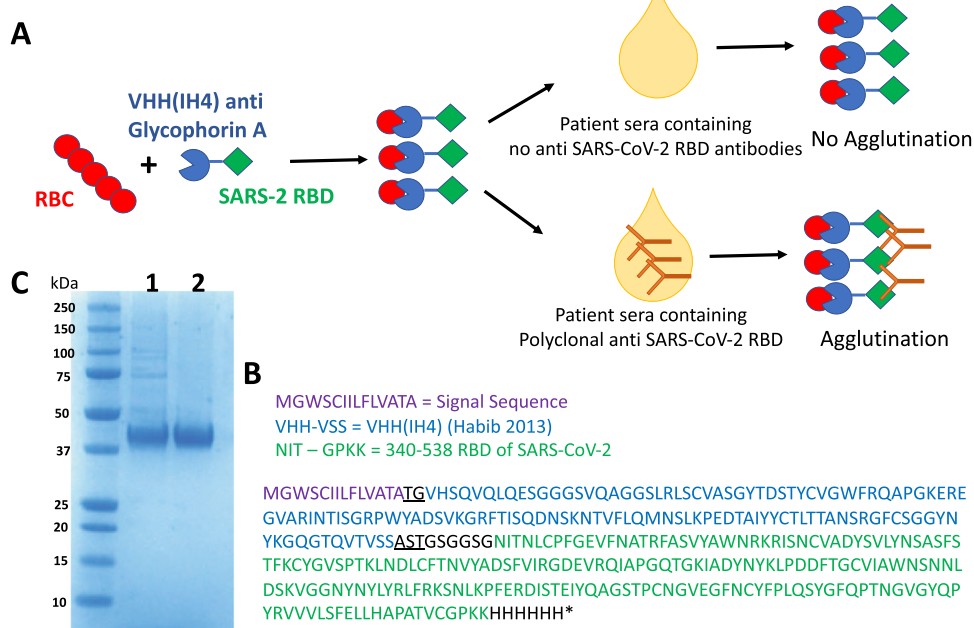

**Fig. 1 Haemagglutination test (HAT) for detection of antibodies to SARS-CoV-2 receptor binding domain. A** Concept of the HAT. **B** Sequence of VHH (IH4)-RBD fusion protein. Residues underlined are encoded by cloning sites AgeI (TG) and SalI (AST). The codon-optimised cDNA sequence is shown in Supplementary Information. **C** SDS-PAGE gel of purified VHH(IH4)-RBD proteins. Three micrograms of protein were run on 4–12% Bolt Bis-Tris under reducing conditions. 1: IH4-RBD produced in house in Expi293F cells. 2: IH4-RBD produced by Absolute Antibody, Oxford, in HEK293 cells. This was done twice with similar results.

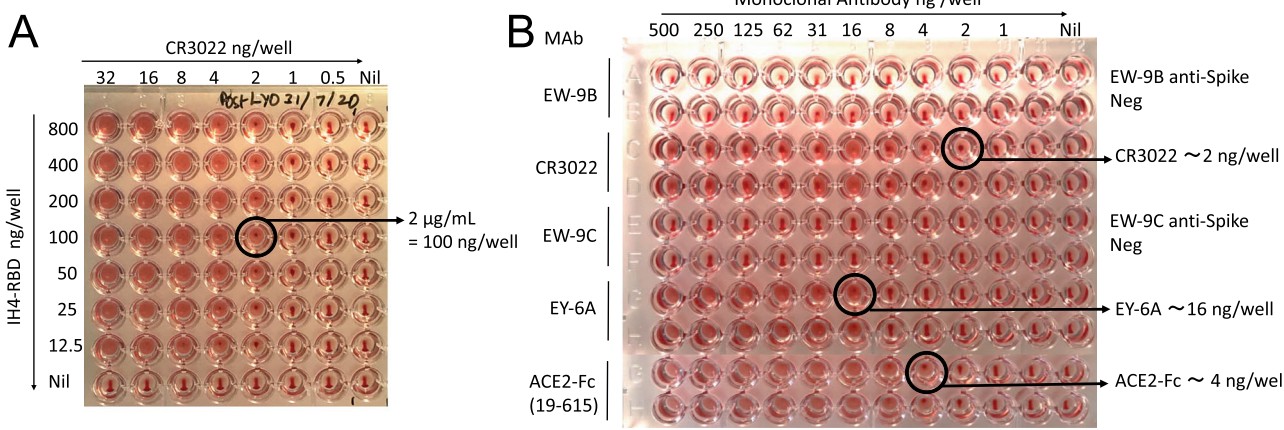

**Fig. 2 Haemagglutination with human monoclonal antibodies or nanobodies to the SARS-CoV-2 RBD. A** Titration of IH4-RBD and monoclonal antibody CR3022 to RBD. Doubling dilutions of CR3022 and IH4-RBD were prepared in separate plates. 50 μL red cells (O−ve whole blood diluted 1:40 in PBS) were added to the CR3022 plate, followed by transfer of 50 μL titrated IH4-RBD. From this titration, 100 ng/well of IH4-RBD was chosen for detection. Similar results were obtained in five other experiments, performed with three separate batches of IH4-RBD. **B** Detection of other anti-RBD monoclonal antibodies and ACE2-Fc. Doubling dilutions of monoclonal antibodies were prepared in duplicate in 50 μL PBS (from a stock solution of 20 μg/mL) from left to right, 50 μL of 1:40 O−ve red cells were added, followed by 50 μL of IH4-RBD (2 μg/mL in PBS). The endpoint was defined as the last dilution without teardrop formation on tilting the plate for ~30 s. The binding sites for CR3022, EY6A, and ACE2 on RBD have been defined[8,10,27,30]. EW-9B and EW-9C are monoclonal antibodies against non-RBD epitopes on the spike protein [28]. ACE2-Fc has been described in ref. [28]. Similar results were obtained in two other titration experiments.

For clinical testing, samples can be supplied as serum or plasma. We therefore compared the effect of serum, K2EDTA, or heparin plasma on the HAT titration endpoints by diluting monoclonal antibodies to each of the four known binding sites on the RBD[20] (FI-3A class 1, C121 class 2, FD-11A class 3, and CR3022 class 4) in serum or plasmas. The titration endpoints obtained in dilutions of serum or plasmas, maintained at 1:20 throughout, did not differ from each other or from the MAbs diluted in PBS (Supplementary Fig. 5). We conclude that the plasma or serum origin of samples is unlikely to influence sensitivity in the haemagglutination assay.

**Agglutination by plasma from donors convalescing from COVID-19.** These experiments established the conditions for detection of haemagglutination by monoclonal antibodies to the

RBD, in particular the optimum concentration of IH4-RBD of 100 ng/well. We then proceeded to look for haemagglutination by characterised plasma from COVID-19 convalescent donors. In the first trial, we tested 18 plasma samples from patients with mostly mild illness, which had been characterised with a quantitative ELISA to detect antibodies to the RBD[32]. For these experiments we used fresh O−ve blood (K2EDTA sample) diluted to 1:40 as a source of red cells to avoid agglutination by natural agglutinins in the plasma. Plasmas were titrated by doubling dilution from 1:20 in 50 µL, then 50 µL of 1:40 O−ve red cells were added, followed by addition of 100 ng of the IH4-RBD in 50 µL PBS. After 1 h incubation, plates were tilted for ~30 s, photographed, and read. The titre of agglutination was assessed by complete loss of teardrop formation by the red cells, any formation of a teardrop was regarded as negative. Figure 3A shows that the HAT titre matched the RBD ELISA results. Four samples were scored as negative in both assays. The remaining results showed that in general the HAT titre increased with the ELISA endpoint titre. One sample gave a positive titre of 1:320 in the HAT but was negative in ELISA (indicated with an arrow). We investigated this sample with a developer composed of the

IH4 nanobody without the RBD component, which revealed that agglutination was RBD dependent (Supplementary Fig. 6). This sample was also positive at 1:1123 in an ELISA for full length spike protein (not shown)[32], which suggests that the antibodies contained in this serum recognised epitope(s) present on the RBD exposed in the HAT, but not on the RBD in the RBD ELISA reference test. The highest titre detected in these samples by the HAT was 1:1280 (Fig. 3B).

We next compared red cell agglutination detected by eye, with detection of binding of monoclonal and serum antibodies to the RBD by FACS analysis. Antibody binding, as detected by indirect immunofluorescence, titrated commensurately with the loss of teardrop formation by the settled red cells, detected by eye (Fig. 3C). Note that full loss of teardrop, our defined objective endpoint, occurred well before saturation of the IH4-RBD labelled red cells with either monoclonal or polyclonal antibodies to the RBD.

Finally, we assessed the degree of variability in the HAT titration of serum/plasma samples introduced by sourcing O−ve indicator red cells from different donors. Four monoclonal antibodies to the four known binding sites on the RBD[20] and four

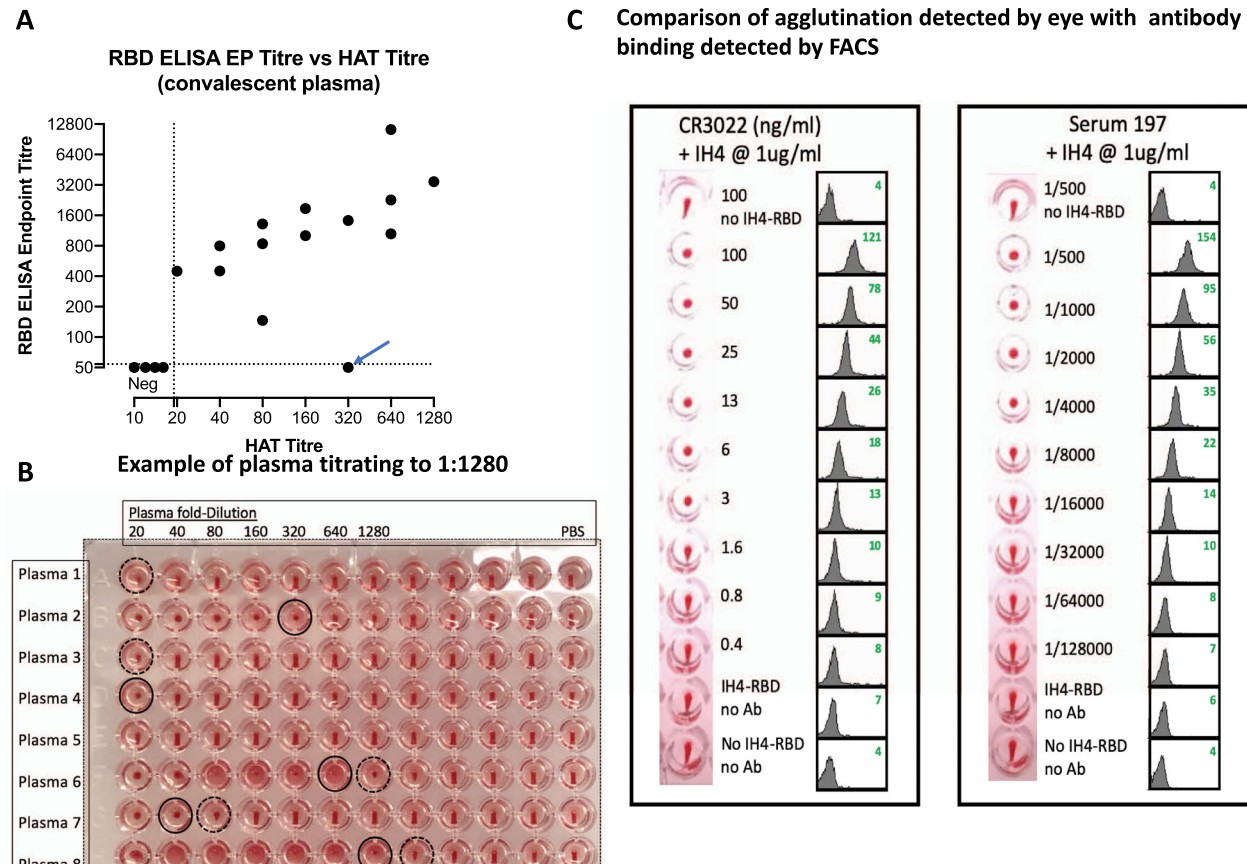

**Fig. 3 Titration of antibodies in the agglutination assay. A** Eighteen plasma samples from mild cases were compared for titration in the HAT with 1:40 O−ve whole blood from a seronegative donor and endpoint titre in an RBD ELISA[32]. Four samples were negative in both assays. The data point marked with an arrow on the graph (plasma 2 on the plate, Fig. 3B) was checked with a reagent composed of IH4 without RBD and shown to be dependent on antibodies to the RBD. This sample did score positive for antibodies to full length spike in an ELISA (endpoint titre 1:1123). These results were confirmed in a repeat assay. **B** An example of titration: positive agglutination endpoints (loss of teardrop) are marked with a black solid-line circle, partial teardrops are marked with a dotted-line circle. **C** Titration of mAb CR3022 and a high titre serum from a COVID-19 patient show that agglutination in the HAT detected by eye correlates with antibody binding to RBCs, as revealed by FACS analysis. Standard HAT titration was performed by double dilutions of RBC suspension containing 1 µg/mL of the IH4-RBD reagent, bar the first and last well. After the HAT assay, the RBCs were stained for FACS analysis by performing three washes before incubation for 60 min at 4 °C with FITC-labelled goat–anti-human IgG. RBCs were then washed twice before analysis by flow cytometry. Green numbers in the upper right corners of the histograms correspond to the geometric mean fluorescence intensity. Original graphs are provided as Supplementary Fig. 4. Similar results were obtained in three other experiments.

sera (selected for high, middle, low, and negative titres) were titrated in quadruplicate on red cells from three donors. The titration endpoints did not vary by more than $+/-1$ doubling dilution between donors. Disagreements between scorers were detected in 5/108 measurements, all within one doubling dilution of each other (Supplementary Tables 3 and 4).

This small amount of variation in endpoints between O−ve red cells sourced from different donors can be controlled by including defined standard sera or monoclonal antibodies in all batches of assays. The WHO/NIBSC (UK) international secondary standard serum 20/130 (recently available from NIBSC UK), which titrates to 1:1280 in the HAT (Supplementary Fig. 3), can be used for calibration of assay endpoints within and between laboratories, and for calibration of local control sera.

These preliminary results showed that the HAT could detect antibodies to the RBD in serum or plasma samples from convalescent patients in a similar manner to an ELISA test, but were not sufficient to establish the sensitivity and specificity of the HAT.

**Sensitivity and specificity of the HAT**. To formally assess the sensitivity and specificity of the HAT, we collected a set of 98 "positive" plasma samples from donors diagnosed with COVID-19 by reverse transcriptase polymerase chain reaction (RT-PCR) at least 28 days prior to sample collection (NHS Blood and Transplant), and 199 "negative" serum samples from healthy donors from the pre COVID-19 era (Oxford Biobank). The samples were randomised before plating. The test wells were arranged in duplicate to contain serum/plasma at 1:40 dilution, and 1:40 O−ve red cells in 50 µL. One hundred nanograms of IH4-RBD in 50 µL PBS was added to one well of the pair, 50 µL of PBS to the other (as a negative control). The negative control is important because in rare cases, particularly in donors who may have received blood transfusions, the sample may contain antibodies to non-ABO or Rhesus D antigens. After development, the plates were photographed, and read by two independent masked observers. Complete loss of teardrop was scored as positive, any flow in the teardrop as negative. These rules were established before setting up the tests. Disagreements (2.3% overall) were resolved by accepting the weaker interpretation—i.e., if one observer scored positive but the other negative, the well was scored as negative. Having completed the scoring the columns of samples were re-randomised and the test and scoring repeated.

Examples of test wells and scoring are shown in Fig. 4. The red cells in all of the negative control (PBS) wells formed a clear teardrop. Red cells in positive wells (indicated with a solid ring) settled either into a mat or a button that failed to form any teardrop on tilting for 30 s. Occasional wells (15 of 297) formed a "partial" teardrop (shown by a dashed ring). These were scored as negative by prior agreement.

With these rules in place, we obtained in the first run sensitivity 88%, specificity 99% and in the second run sensitivity 93%, specificity 99% (Fig. 4). The Siemens Atellica Chemiluminescence assay for detection of IgG antibodies to the RBD was run in parallel on 293 of these 297 samples and gave sensitivity 100%, specificity 100% for this sample set.

We decided prior to this formal assessment to score wells with partial teardrop formation as negative, as these wells tended to give rise to disagreements between scorers and were associated with very low-level of staining in the FACS analysis (Fig. 3C). Overall, 15 of 297 wells gave a partial teardrop. Six of these fifteen were from PCR −ve donors and scored negative on the Siemens assay, and nine out of sixteen were from PCR+ve donors and were Siemens positive. If partial teardrops were scored as positive, the sensitivity of the two assays increased to 97 and 99% (from 88 and 93%), but specificity

was reduced to 96 and 98% (from 99%). This small loss of specificity would be unacceptable in serosurveys where the expected prevalence of previous SARS-CoV-2 infection was low.

**HAT in the hospital setting**. We next assessed the HAT in the setting of patients recently admitted to hospital (the first 5 days) through access to the COMBAT collection of samples (see "Methods"). This set comprised 153 plasma samples from donors diagnosed with COVID-19 by PCR, with clinical syndromes classified as « Critical », « Severe », « Mild », and «PCR positive Health Care Workers ». Seventy-nine control plasma samples donated in the pre COVID-19 era were obtained either from patients with bacterial sepsis (54 samples) or healthy volunteers (25 samples). Samples were titrated in 11 doubling dilutions of 50 µL from 1:40–1:40,096 (columns 1–11). Column 12 contained 50 µL PBS as a negative control. Fifty microlitres of 1:40 O−ve whole blood was added, followed by 50 µL of 2 µg/mL IH4-RBD (100 ng/well). In parallel, all of the 153 samples from PCR positive donors were assessed by the Siemens Atellica Chemiluminescence test for antibodies to the RBD of the spike protein. The HAT scores (as the number of doubling dilutions of the sample required to reach the endpoint of complete loss of teardrop), and representative agglutination results are shown in Fig. 5A. In Fig. 5B, the HAT scores are plotted with their related Siemens test scores.

None of the seventy-nine negative control samples scored as positive in the HAT at a dilution of 1:40, thus providing 100% specificity in this set of samples. The HAT detected 131/153 (86% sensitivity) of the samples from PCR-diagnosed donors within the first 5 days of hospital admission, whereas the Siemens test detected 113/153 (74%). On day 5 the HAT detected 41/45 (91% sensitivity). Two samples had an endpoint greater than 11 doubling dilutions in the HAT, and required a repeat measurement spanning two plates. These two samples titrated to 13 doubling dilutions (1:163,840). Unmasking the samples revealed that both were acquired from an elderly lady with mild disease on days 3 and 5 of her admission. The range of positive titres detected by HAT was broad: 1–13 doubling dilutions (1:40–1:163,840). A correlation coefficient with the Siemens test could not be calculated as the latter has a ceiling score of 10 (Fig. 5B). A comparison of the two tests in a contingency table with cut-off of 1:40 (first doubling dilution) for HAT, and a score $\geq 1$ for the Siemens test (as defined by the manufacturer), showed a strong correlation between the two tests for detection of antibodies to the RBD ($P < 0.0001$; two-tailed Fisher's exact test, Fig. 5B). Overall, 52 of the 153 samples were from 24 donors with COVID-19 from whom repeated samples were taken on days 1, 3, or 5 of admission. The HAT detected a rise in agglutination titre over the first 5 days of admission in 16/24 (67%) of these patients (Table 1, Repeat in Supplementary Table 2). Reductions in titre were not detected.

These results showed that in the setting of hospital admission in the UK for suspected COVID-19 disease, the HAT has an overall sensitivity of 86% and specificity of 100% by day 5, and frequently (67%) detected a rise in HAT titre during the first 5 days of admission. In this context, the HAT performed at least as well as the commercially available Siemens Atellica Chemiluminescence assay (74%) for the detection of antibodies to the RBD of SARS-CoV-2 spike protein. Twenty samples were negative in both tests, but nine of these were taken on day 1 of admission, which suggests that both tests have lower detection levels early in the course of hospital admission, before the antibody response has fully developed.

The O−ve blood used as indicator for this experiment was collected into a heparin tube, and then transferred to a K2EDTA

## A Examples from screen of 297 test samples

## B Operating Characteristics of the Haemagglutination Test

98 PCR +ve Samples >=28 days post diagnosis
199 −ve samples from The Oxford Biobank Pre Covid-19

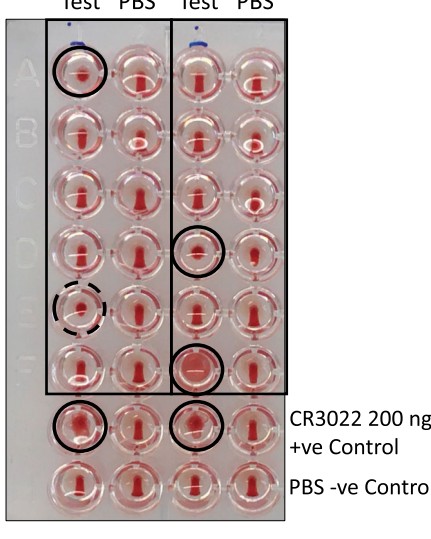

Test  PBS  Test  PBS

CR3022 200 ng +ve Control

PBS -ve Control

| 11 June | | | |
|---|---|---|---|
| **Sensitivity and Specificity Calculator** | | | |
| +VE = No Teardrop | | | |
| | HAT +VE | HAT -VE | |
| PCR +VE | 86 | 12 | 98 |
| PCR -VE | 1 | 198 | 199 |
| | 87 | 210 | 297 |
| | | | |
| Sensitivity % | 88 | | |
| Specificity % | 99 | | |
| Likelihood Ratio | 174.6 | | |
| 13 June | REPEAT | | |
| **Sensitivity and Specificity Calculator** | | | |
| +VE = No Teardrop | | | |
| | HAT +VE | HAT -VE | |
| PCR +VE | 91 | 7 | 98 |
| PCR -VE | 1 | 198 | 199 |
| | 92 | 205 | 297 |
| | | | |
| Sensitivity % | 93 | | |
| Specificity % | 99 | | |
| Likelihood Ratio | 184.8 | | |

**Fig. 4 Operating characteristics of the HAT. A** The test set of 297 randomised plasma samples was diluted 1:40 and mixed with 1:40 O−ve blood in two columns. IH4-RBD (100 ng in 50 μL) was added to the test samples and PBS to negative control wells. The plates were incubated at room temperature for 1 h to allow the red cell pellet to form, then tilted for ~30 s to allow a teardrop to form. Complete loss of teardrop was scored as positive agglutination (marked with a black solid-line circle). Full teardrop or partial teardrop (marked with a dotted-line circle) was scored negative. The samples in columns were re-randomised and tested a second time 2 days later. **B** Contingency table showing the operating characteristics of the HAT. Source data are provided as a Source Data file.

tube. In order to be sure that the presence of heparin in the red cells had not altered the behaviour of the test, and to confirm the robustness of the results, we repeated the titrations on all of the 232 samples 34 days later, with fresh O−ve red cells from a different donor collected as usual into a K2EDTA tube. The results are shown in Supplementary Fig. 2A–C. Specificity of the HAT remained at 100% (none of the 79 control samples were detected as positive at 1:40). The correlation with the previous assay was strong ($R^2 = 0.975$), and 99% of the 232 titrations were within one doubling dilution of the matched earlier measurement. The slope of the correlation was 0.94 (95% CI 0.92–0.96), significantly less than 1. This was due to a proportion of results titrating to one doubling dilution lower titre. However, this had only a small impact on sensitivity (81% from 86%), which was still an improvement on the Siemens test (74%) in this context of the first 5 days of hospital admission. A rise in HAT titre in 16/24 during the first 5 days of admission was confirmed.

**HAT as a point of care test on capillary samples**. The HAT is designed to detect antibodies to the RBD starting at a serum dilution of 1:40, and we have found that 50 μL of 1:40 dilution of whole blood provides an optimal concentration of red blood cells for detection by agglutination in V-bottomed 96-well plates. We have not completed an extensive analysis of the HAT as a point of care test. However, we have preliminary evidence that lyophilised IH4-RBD sent to the National Institute of Immunology, New Delhi, functions as a point of care test on capillary blood obtained by finger prick. In Fig. 6, three positive (donors 1, 2, and 3) and three negative (donors 4, 5, and 6) HAT results are compared to a standard ELISA for detection of antibodies to the RBD.

Although our results clearly show that HAT represents a simple, sensitive, robust, and very inexpensive test to detect antibodies against RBD, further work will be needed to establish the operating characteristics of the HAT as a point of care test on capillary samples. As a starting point, we provide a suggested operating procedure for capillary samples in "Methods".

**Distribution of the IH4-RBD as lyophilised protein**. In order to ship the IH4-RBD reagent efficiently, we have examined the effects of lyophilisation and reconstitution with water. IH4-RBD synthesised for the purpose of distribution was provided at 5 mg/mL in PBS by Absolute Antibody Ltd, Oxford. Two hundred microlitre aliquots (1 mg, enough for 10,000 test wells) were lyophilised overnight and stored at −20 °C. Aliquots were thawed, reconstituted with 0.2 mL of double distilled water, diluted to 1 mg/mL with 0.8 mL of PBS, and titrated against the pre-lyophilisation material. No change in the titration occurred. We have synthesised 1 g of IH4-RBD, sufficient for 10 million test wells. This is available free of charge for any qualified group anywhere in the world in aliquots of 1 mg (10,000 test wells).

## Discussion

The COVID-19 pandemic has had a particularly gruelling influence on the world economy and on most populations of the world. The appearance of such a new highly contagious virus will probably not be a unique occurrence in the decades ahead. One of the lessons learned is the importance of developing affordable serological tests for detection of immune responses to SARS-CoV-2. Commercial antibody tests are not widely available to low- and middle-income countries, and lateral flow assays, while offering early promise as a near-patient test, have failed to deliver in terms of performance metrics, are expensive, and there are concerns about significant batch-to-batch variation[33]. By contrast, the advantages of the HAT are the low cost of production of its single reagent (~0.27 UK pence per test well, and altogether less than 1 £/test if the costs of other consumables are taken into

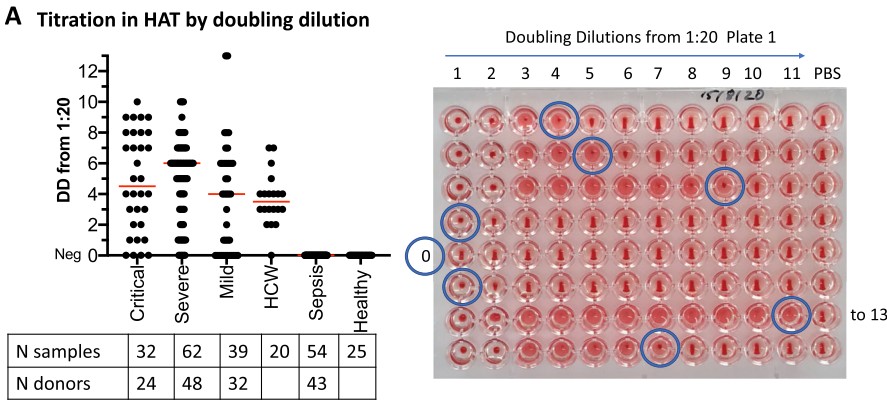

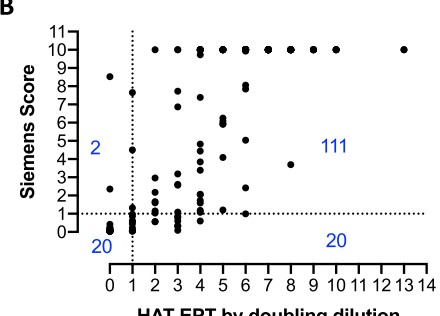

**A** Titration in HAT by doubling dilution

Doubling Dilutions from 1:20 Plate 1

| N samples | 32 | 62 | 39 | 20 | 54 | 25 |
|-----------|-----|-----|-----|-----|-----|-----|
| N donors | 24 | 48 | 32 | | 43 | |

Donors were grouped according to clinical criteria; HCW = Front Line Health Care Workers; Sepsis = Samples with patients with Sepsis prior to Covid-19 pandemic; Healthy = Healthy volunteers prior to Covid-19; DD 1 = 1:40. Actual Titre = 20 x 2^DD.

**B**

| | HAT+ | HAT- | Total | Fraction |
|-----------|------|------|-------|----------|
| Siemens + | 111 | 2 | 113 | 0.98 |
| Siemens - | 20 | 20 | 40 | 0.50 |
| Total | 131 | 22 | 153 | 0.86 |
| Fraction | 0.85 | 0.91 | 0.74 | |

**P value and statistical significance**

| Test | Fisher's exact test |
|------|---------------------|
| P value | <0.0001 |
| P value summary | **** |
| One- or two-sided | Two-sided |

Comments
Of 22 Samples −ve in HAT, 10 were collected on Day 1
Of 20 samples −ve in both assays, 9/20 were collected on Day 1

**Fig. 5 Titration of the set of 232 samples in the HAT. A** The collection included 32 samples from 24 critical patients, 62 samples from 48 severe, 39 samples from 32 mild, 20 single samples from healthcare workers (HCW), 54 samples from 43 patients with unrelated sepsis in the pre COVID-19 era, and 25 samples from healthy unexposed controls. Median is indicated by a red line. DD doubling dilutions. **B** Comparison to Siemens result (anti-RBD) with HAT titre by doubling dilution for 153 samples from critical, severe, mild, and HCW SARS-CoV-2 PCR positive donors. Source data are provided as a Source Data file.

account), better performance than most lateral flow devices[33], and versatility in not requiring anything other than a source of O −ve blood (10 mL of K2EDTA blood provides enough for 8000 tests), an adjustable pipette, PBS, and a standard 96-well V-bottomed plate.

We have demonstrated that the HAT functions as a viable test for the presence of antibodies to the RBD of the SARS-CoV-2 spike protein in stored serum/plasma samples, using O−ve red cells as indicators. In the formal assessment of sensitivity and specificity, we recorded an average 90% sensitivity and 99% specificity (likelihood ratio ~175), compared to 100% sensitivity and 100% specificity for the Siemens Atellica Chemiluminescence test on the same set of samples. The "positive" samples were selected to have been taken at least 28 days after a positive PCR test. These conditions are optimal for serological tests by allowing time for a rise in antibodies. The sensitivity of ~90% and specificity of 99% did not reach the level of 98% for both recommended by the UK MHRA[19] or 99.5% for both recommended by the Infectious Disease Society of America[34]. However, these values are still consistent with a useful test in appropriate contexts, provided that users are fully aware of the operating characteristics and interpret the results correctly. Our comparison of the HA test with binding detected by FACS analysis suggests that samples that fail to be detected by agglutination have very low titre.

The sensitivity of the single point HAT can be enhanced (to ~98%) if wells with partial teardrop formation are scored as positive. However, this improvement in sensitivity is gained at the expense of a reduced specificity (to ~97%). If partial teardrops are

to be scored as positive in the spot test at 1:40, we recommend obtaining confirmation for these by ELISA. Improvement in the operating characteristics of the HAT may be possible by a systematic analysis of buffer composition and experimental conditions, and is being investigated. We have deliberately kept complexity to a minimum, and thus all dilutions were made in standard PBS, and for the moment we recommend scoring wells with partial teardrop formation as negative.

Titration of the HAT provides additional quantitative information. Titrations repeated a month apart are highly reproducible (Supplementary Fig. 2). Minor variations in endpoints (+/− 1 doubling dilution) are detected between assays performed with O−ve red cells from different donors (Supplementary Tables 3 and 4). However, assays can be calibrated within and between laboratories by inclusion of the WHO standard serum 20/130 now available from NIBSC UK (which titrates to 1:1280 in the HAT), and monoclonal antibodies with defined endpoints such as CR3022.

It is interesting that the HAT titrations actually performed a little better than the Siemens test on 153 stored plasma samples from donors during the first 5 days of their hospital admission (note that symptom onset may have been several days earlier), in whom it detected 86% (81% in the repeat) of samples from PCR-diagnosed donors, compared to 74% for the Siemens test, and gave 100% specificity for the sample set containing control plasma from patients with sepsis and healthy controls. We speculate that at this early period of the COVID-19 illness, the immune response may be dominated by IgM that would be expected to be particularly efficient at crosslinking the IH4-RBD

labelled red cells. In addition, of the 24 donors who were tested more than once in the first 5 days in hospital, the HAT detected a rise in titre in 16 (67%). A fixed high titre was detected in a further 4 to provide a sensitivity of 83% in these 24 cases. In situations of high clinical suspicion, the HAT could potentially have a place as a helpful test to support the diagnosis of COVID-19 by detecting a rising titre of antibodies to the RBD during hospital admission. In patients with a prior probability of a diagnosis of COVID-19 of ~10%, the likelihood ratio of ~175 for the HAT provides a posterior probability of ~95% for this diagnosis. However, it is essential that if clinicians use a rising titre in the HAT as a diagnostic aid, they should be aware of the relatively low sensitivity (~67%) in this context.

Finally, we show that the lyophilised IH4-RBD reagent sent to New Delhi functioned as expected in preliminary point of care testing on capillary samples obtained by finger prick. However, additional evidence is needed to show that the sensitivity and specificity of the HAT, applied as a point of care test in this way, are comparable to the tests on stored plasma samples, as stressed by the IDSA guideline on serological testing[34]. This will need to be done in field conditions, which is planned.

In the absence of knowledge about the level of antibody that indicates protection, the HAT should not be used to provide personal results to individuals, as discussed by the UK Royal College of Pathologists (https://www.rcpath.org/profession/on-the-agenda/COVID-19-testing-a-national-strategy.html). It should also not be used as a diagnostic test in clinical practice as it is not yet validated for this purpose. It should be most useful as an inexpensive quantitative research method to follow seropositive individuals post infection or post vaccination.

The technique required for applying the HAT can be learned in a day by a trained laboratory technician, paramedic, nurse, or doctor. We have produced 1 g of the developing IH4-RBD reagent (enough for ten million test wells) and offer to ship lyophilised aliquots of this material (sufficient for 10,000 tests) anywhere in the world, free of charge, for use as a research reagent for serological studies of COVID-19.

## Methods

**Sample collection and ethics**. Figures 1 and 2: Control whole blood (K2EDTA) as a source of red cells was collected from a healthy donor after informed consent. The use was approved by HTA license 12433.

Figure 3: Pre pandemic negative controls: these samples were collected from healthy adults in the Oxfordshire region of the UK between 2014 and 2016, ethics approval: Oxfordshire Clinical Research Ethics Committee 08/H0606/107+5. Positive sample set: these were convalescent plasma donors recruited by NHS Blood and Transplant (NHSBT), ethics approval (NHSBT; RECOVERY [Cambridge East REC (ref: 20/EE/0101)] and REMAP-CAP [EudraCT 2015-002340-14] studies). The serum from a convalescent patient used for FACS analysis was left-over clinical material obtained from the Virology Laboratory of the Toulouse Hospital, where, by default, all patients gave informed consent for such materials to be used for research purposes.

**Table 1 Fifty-two samples from 24 donors who were sampled repeatedly during the first 5 days in hospital.**

| | | Day post hospitalisation | | |
|---|---|---|---|---|
| No. | Case | 1 | 3 | 5 |
| **1** | **C2** | | **40** | **160** |
| **2** | **C6** | | **40** | **80** |
| 3 | C7 | 0 | | 0 |
| **4** | **C8** | | **320** | **1280** |
| **5** | **C9** | **40** | | **2560** |
| 6 | C10 | 320 | 320 | |
| **7** | **C23** | **0** | **160** | **640** |
| **8** | **S5** | **80** | **160** | |
| **9** | **S6** | **1280** | **1280** | **2560** |
| **10** | **S7** | **80** | | **2560** |
| **11** | **S12** | **640** | **1280** | |
| **12** | **S13** | **0** | **40** | **320** |
| 13 | S14 | | 640 | 640 |
| 14 | S17 | | 2560 | 2560 |
| **15** | **S20** | **160** | **320** | |
| **16** | **S31** | **40** | | **1280** |
| **17** | **S41** | | **640** | **1280** |
| **18** | **S43** | | **40** | **160** |
| **19** | **S48** | | **1280** | **2560** |
| **20** | **M9** | | **0** | **160** |
| 21 | M11 | 0 | 0 | 0 |
| 22 | M13 | | 0 | 0 |
| 23 | M14 | | 163840 | 163840 |
| **24** | **M25** | **40** | | **640** |
| 25 | M30 | | 320 | 320 |

Numbers in bold indicate rising titres.
C critical, S severe, M mild.

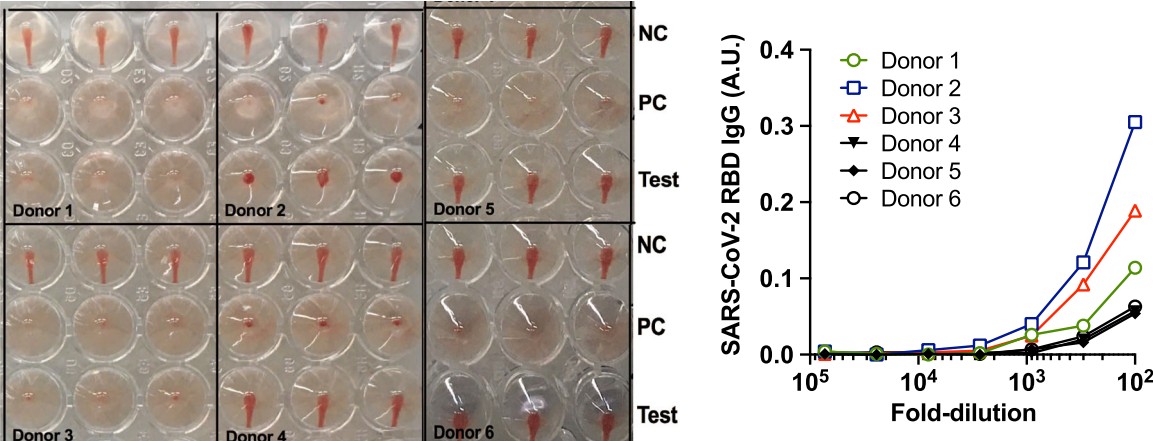

**Fig. 6 HAT as a point of care test.** Capillary blood samples were obtained by Lancet. Antibodies to the RBD were detected by HAT on autologous red cells in the sample in "Test" wells (plasma at 1:40) after addition of 100 ng/well IH4-RBD (see "Methods"). NC negative control (PBS replaces IH4-RBD), PC positive control (20 ng/well CR3022, an anti-RBD monoclonal antibody added). In parallel, after removal of red cells, the plasma was tested in a standard ELISA for detection of antibodies to the RBD. Low levels of antibody detected in the ELISA were sufficient to give a positive result in the HAT. Detection of antibodies by HAT on autologous red cells obtained by finger prick has been repeated multiple times in both Oxford and Toulouse, and will be the subject of a future report. Source data are provided as a Source Data file.

Figures 4 and 5 and Table 1: Known COVID-19 positive samples were collated from three ethically approved studies: Gastro-intestinal illness in Oxford: COVID substudy [Sheffield REC, reference: 16/YH/0247]ISARIC/WHO, Clinical Characterisation Protocol for Severe Emerging Infections [Oxford REC C, reference 13/SC/0149], the Sepsis Immunomics project [Oxford REC C, reference:19/SC/0296]), and by the Scotland A Research Ethics Committee (Ref: 20/SS/0028). Patients were recruited from the John Radcliffe Hospital in Oxford, UK, between March and May 2020 by identification of patients hospitalised during the SARS-COV-2 pandemic and recruited into the Sepsis Immunomics and ISARIC Clinical Characterisation Protocols. Time between onset of symptoms and sampling were known for all patients and if labelled as convalescent patients were sampled at least 28 days from the start of their symptoms. Written informed consent was obtained from all patients. All patients were confirmed to have a test positive for SARS-CoV-2 using RT-PCR from an upper respiratory tract (nose/throat) swab tested in accredited laboratories. The degree of severity was identified as mild, severe, or critical infection according to the recommendations from the World Health Organisation. Severe infection was defined as COVID-19 confirmed patients with one of the following conditions: respiratory distress with RR > 30/min; blood oxygen saturation < 93%; arterial oxygen partial pressure (PaO$_2$)/ fraction of inspired O2 (FiO$_2$) < 300 mmHg; and critical infection was defined as respiratory failure requiring mechanical ventilation or shock or other organ failures requiring admission to ICU. Comparator samples from healthcare workers with confirmed SARS-CoV-2 infection who all had mild non-hospitalised disease were collected under the gastro-intestinal illness in Oxford: COVID substudy and samples from patients with equivalently severe disease from non-COVID infection were available from the Sepsis Immunomics study where patients presenting with significantly abnormal physiological markers in the pre pandemic timeframe had samples collected using the same methodology as that applied during the COVID pandemic. Blood samples were collected in K2EDTA vacutainers and PBMCs were separated from plasma using Sepmate isolation tubes (STEMCELL Technologies) and plasma was used in the downstream HAT assay.

Figure 6: Capillary samples were collected from members of institute staff with informed consent (New Delhi, India). The study is a part of the COVID-19 project "IPA/2020/000077". The project has been approved by the Institutional Human Ethics Committee; Ref. no.–IHEC#128/20.

**Cloning, expression, and purification of VHH(IH4)-RBD**. The amino acid sequence for IH4 was obtained from US Patent No US 9,879,090 B2, kindly corrected by Olivier Bertrand. The codon-optimised gene encoding IH4-RBD sequence (Fig. 1B and supplementary for the cDNA sequence) was synthesized by Integrated DNA Technologies. The gene was cloned into the AbVec plasmid (Genbank FJ475055) using the restriction sites AgeI and HindIII (the vector supplied the signal sequence). This expression plasmid for IH4-RBD is available on request. Protein was expressed in Expi293F$^{TM}$ cells using the manufacturer's protocol (Thermo Fisher). Protein supernatant was harvested on day 5/6 after transfection, spun, and 0.22 μm filtered. The protein was affinity-purified using a His-Trap HP column (Cytiva). Binding buffer consisted of 20 mM sodium phosphate, 150 mM NaCl, and 20 mM imidazole at pH 7.4 and the elution buffer of 500 mM imidazole in 1x binding buffer. Protein was concentrated using 15 mL Vivaspin 30 kDa MWCO filter and then buffer exchanged to PBS using a 10 mL 7 kDa Zeba spin column (Thermo Fisher).

For large-scale production, the protein was synthesized by Absolute Antibody Ltd, Oxford, using the same plasmid construct in HEK293 cells.

**Monoclonal antibodies**. The variable heavy and light genes of the monoclonal antibodies CR3022[26], VHH72-Fc[31], C121[21], H11-H4-Fc[29], and S309[24] were synthesized by Integrated DNA Technologies IDT and cloned into antibody expression vectors (GenBank FJ475055 and FJ475056). Antibody was expressed using the ExpiCHO-S expression system according to the manufacturer's protocol. Antibodies EY6A, FI-4A, FI-3A, FD-5D, FD-11A, FN-12A, FJ-10B, FM-7B, EW-8B, EW-9C, and FJ-1C were developed in our laboratory[28].

**Lyophilisation of IH4-RBD and CR3022 monoclonal antibody**. For lyophilisation, 200 μL (1 mg) of IH4-RBD (5 mg/mL) and 100 μL (200 μg) CR3022 mAb (2 mg/mL) in PBS buffer prepared in Protein Lo-Bind microcentrifuge tube (Fisher Scientific) were frozen at −80 °C and further cooled down to −196 °C using liquid nitrogen. Precooled samples were transferred to BenchTop K freeze dryer (VirTis) with chamber at 49 μbar and condenser precooled to −72.5 °C. The samples were freeze-dried for a minimum of 24 h, wrapped in Parafilm (Merck), and stored at −20 °C. Lyophilised sample was reconstituted in the same original volume of MilliQ water: 200 μL for IH4-RBD and 100 μL for the CR3022. The IH4-RBD was then diluted to 1 mg/mL stock solution by addition of 800 μL of sterile PBS and stored at 4 °C. The CR3022 was diluted 1:100 to 20 μg/mL in PBS on the day of experiment for standard titrations.

**Indirect Immunofluorescence of RBD labelled red cells (Fig. 3C)**. Titrations of Mab CR3022 and a positive serum by combined HAT and FACS analysis: an EDTA blood sample from an O−ve donor was obtained from the Toulouse blood bank, and a 1/40 dilution in PBS was used for the assay. Serum 197, from a

convalescent COVID-19 patient, was kindly provided by Laurence Abravanel and Jacques Izopet (Virology service, Toulouse Hospital). The HAT was performed in 100 μL/well, and pictures were taken after 60 min incubation at room temperature and 30 s tilt of the plate. Red blood cells were then washed thrice in PFN (PBS + 2% foetal calf serum + azide 0.2 g/L), before staining with a goat anti-human IgGAM-FITC polyclonal secondary antibody (Jackson, Cat. No. 109-095-064, used 1/200) for 60 min on ice. Cells were then washed twice more in PFN before analysis on a FACScalibur flow cytometer, using the Cellquest Pro software 5.2 to calculate the mean geometric fluorescence intensity on 5000 cells within a broad FSC/SSC rectangular gate (see Supplementary Fig. 4).

**Indirect ELISA to detect SARS-CoV-2 specific IgG (Fig. 6)**. A standard indirect ELISA was used to determine the SARS-CoV-2 specific IgG levels in plasma samples. A highly purified RBD protein from SARS-CoV-2 Wuhan strain (NR-52306, BEI Resources, USA), expressed in mammalian cells, was used to capture IgG in the plasma samples. Briefly, ELISA plates (Nunc, MaxiSorp) were coated with 100 μL/well of RBD antigen diluted in PBS (pH 7.4) at the final concentration of 1 μg/mL and incubated overnight at 4 °C. Plates were washed three times with washing buffer (0.05% Tween-20 in PBS) followed by the incubation with blocking buffer (3% skim milk and 0.05% Tween-20 in PBS). The threefold serially diluted heat inactivated plasma samples in dilution buffer (1% skim milk and 0.05% Tween-20 in PBS) were added into the respective wells, followed by incubation at room temperature for 1 h. After incubation, plates were washed, and anti-human IgG conjugated with horseradish peroxidase (Southern Biotech) was added in each well. After 1 h incubation, plates were washed and developed by OPD-substrate (Sigma-Aldrich) in dark at room temperature. The reaction was stopped using 2N HCl and the optical density (OD) was measured at 492 nm. The RBD-antigen coated wells that were added with sample diluent alone were used as the blank. The OD values from sample wells were plotted after subtracting the mean of OD values obtained in the blank wells.

**HAT protocol**. Step by step protocol for HAT is available on Protocol Exchange (https://doi.org/10.21203/rs.3.pex-1367/v1)[35].

Equipment and reagents for HAT

(1) O−ve blood as a source of red cells collected in K2EDTA tube, diluted in PBS to 1:20 or 1:40 as needed. Resuspend by inverting gently ~12 times.
(2) BD Contact Activated Lancet Cat. No. 366594 (2 mm × 1.5 mm).
(3) 100 μL, 20 μL pipettes, multichannel pipettes.
(4) V-bottomed 96-well plates (Greiner Bio-One, Cat. No. 651101, Microplate 96-well, PS, V-bottom, Clear, ten pieces/bag).
(5) Eppendorf tubes.
(6) K2EDTA solution (add 5 mL PBS to 10 mL K2EDTA blood collection tube = 3.6 mg K2EDTA/mL, store at 4 °C).
(7) PBS tablets (OXOID Cat. No. BR0014G).
(8) IH4-RBD Reagent diluted 2 μg/mL in PBS. This remains active for at least 1–2 weeks stored at 4 °C.
(9) V-bottomed 96-well plates, numbered, dated, and timed (helps when timing many plates).
(10) Positive control monoclonal antibody CR3022 diluted to 2 μg/mL in PBS.

Other reagents

Monoclonal antibody to human IgG (gamma chain specific) Clone GG-5 Sigma Cat. No. I5885.

(1) Spot test on stored serum/plasma samples (Fig. 4).

 (1) Plate out 50 μL of 1:20 serum/plasma in alternate columns 1, 3, 5, 7, 9, and 11 (add 2.5 μL sample to 47.5 μL PBS).
 (2) Add 50 μL 1:20 O−ve blood collected (so that now sample is diluted to 1:40 and red cells at ~1% v/v).
 (3) Mix and transfer 50/100 μL to neighbouring columns 2, 4, 6, 8, 10, and 12 for −ve controls. The negative control is important because in rare cases, particularly in donors who have received blood transfusions, the sample in principle may contain antibodies to non-ABO or Rhesus D antigens.
 (4) Add 50 μL IH4-RBD reagent (2 μg/mL in PBS = 100 ng/well) to columns 1, 3, 5, 7, 9, and 11.
 (5) Add 50 μL PBS to columns 2, 4, 6, 8, 10, and 12.
 (6) Inc 1 h RT.
 (7) Tilt for 30 s.
 (8) Photograph: with mobile phone use the zoom function to obtain a complete field.
 (9) Read as Positive = No teardrop, Negative < 1:40 = partial teardrop, Neg = complete teardrop.
 (10) Two readers should read the plates independently, and disagreements resolved by taking the lesser reading.
 (11) For each batch of samples set up positive control wells containing 20–100 ng monoclonal antibody CR3022 (as in finger-prick test below). This establishes that all of the reagents are working.

(2) Titration of stored serum/plasma samples.

(1) Dilute samples to 1:20 in 50 μL PBS (2.5–47.5 μL) in V-bottomed plate in rows A–H, column 1. Prepare WHO standard serum 20/130 as above at 1:20 or CR3022 50 μL at 20 μg/mL, for calibration for each batch of titrations. WHO standard 20/130 should titrate to ~1:1280 and 20 μg/mL CR3022 to ~1:512.

(2) Prepare doubling dilutions with PBS across the plate columns 1–11 (1:40–1:40,960), PBS control in column 12. Eight samples can be titrated per 96-well plate.

(3) Add 50 μL 1:40 O−ve red cells (1% v/v or 1:40 fresh EDTA O−ve blood sample) to all wells.

(4) Add 50 μL IH4-RBD (2 μg/mL, =100 ng/well). [Note: the red cells and IH4-RBD can be premixed and added together in either 50 or 100 μL volume, to save a step. This variation in technique does not alter the measured titres.]

(5) Allow red cells to settle for 1 h.

(6) Tilt plate for at least 30 s and photograph. The titre is defined by the last well in which the teardrop fails to form. Partial teardrop regarded as negative.

(3) Finger-prick test on capillary blood as a point of care test.

(1) Preparation: clean hands, warm digit. Prepare a plate (96-well V-bottomed) labelled with date and time.

(2) Prick skin on outer finger pulp with disposable, single use BD or another Lancet.

(3) Wipe away first drop of blood with sterile towel/swab.

(4) Massage second drop.

(5) Take a minimum of 5 μL blood with 20 μL pipette, mix immediately into 20 μL K2EDTA (3.6 mg/mL/PBS) in Eppendorf. If possible, take 25 μL of blood and mix into 100 μL K2EDTA solution. Another approach is collection of blood drops into a BD Microtainer K2E EDTA lavender vials REF 365975 that take 250–500 μL.

(6) For 5 μL sample dilute to 200 μL with PBS (add 175 μL PBS), for 25 μL sample dilute to 1 mL (add 975 μL PBS). Sample is now at 1:40, and the red cells are at the correct density (~1% v/v assuming a haematocrit of 40%) to give a clear teardrop.

(7) Plate 50 μL × 3 in V-bottomed microtitre wells labelled T (Test), + (PC, positive control), and − (NC, negative control).

(8) Add 10 μL of control anti-RBD Mab CR3022 (2 μg/mL stock in PBS, 20 ng/well) to "+" well.

(9) Add 50 μL IH4-RBD (2 μg/mL in PBS) to "T" (Test) and "+ve" wells, 50 μL PBS to "−ve" well.

(10) Incubate 1 h at RT for red cells to form a pellet in the "−ve" well.

(11) Tilt plate against a well-lit white background for ~30 s to allow teardrop to form in "−ve" well.

(12) The presence of antibodies to RBD is shown by loss of teardrop formation in the "T" and "+ve" wells. Occasionally, a partial teardrop forms—these wells are counted as negative.

(13) Photograph the plate to record the results with the date and time. Results can be reviewed and tabulated later. Taking picture from a distance and using the zoom function helps to take a clear picture of all wells in a 96-well plate.

(14) The negative (PBS) control should be done on every sample for comparison. The Positive control induced by CR3022 is used to check that all the reagents are working, and that the glycophorin epitope recognised by VHH(IH4) is present on the red cells. Absence of the IH4 epitope should be very rare 6. For setting up cohorts a positive control on every sample is therefore not necessary but should be included in every batch of samples.

(15) If a 25 μL sample of blood was taken from the finger prick there should be 850 μL of the 1:40 diluted blood left. The red cells can be removed and a preparation of 1:40 O−ve red cells used as above to titrate the sample. In principle, the autologous red cells could be washed ×3, resuspended in the same volume of PBS, and used as indicators for the titration, however we have not attempted to do this. The supernatant is 1:40 plasma that can be used in confirmatory ELISA or other tests.

**Reporting summary**. Further information on research design is available in the Nature Research Reporting Summary linked to this article.

## Data availability

All the data that support the findings of this study are available in the accompanying Source Data file. Any other relevant data are available from the authors upon reasonable request.

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

## Acknowledgements

A.T. and P.R. are funded by the Medical Research Council (MR/P021336/1), Townsend-Jeanet Charitable Trust (charity number 1011770), and the Chinese Academy of Medical Sciences (CAMS) Innovation Fund for Medical Science (CIFMS), China (grant no. 2018-I2M-2-002). T.K.T is funded by the EPA Cephalosporin Early Career Research Fellowship and Townsend-Jeanet Charitable Trust (charity number 1011770). N.G. is funded by the Science Engineering Research Board, Department of Science and Technology, India. P.C.M. is funded by the Wellcome Trust (grant ref 110110Z/15/Z). D.J.R., D.C., T.S. and S.H. are supported by NIHR Oxford Biomedical Research Centre. G.R.S. was supported by the National Institute for Health Research Biomedical Research Centre Funding Scheme, the Chinese Academy of Medical Sciences (CAMS) Innovation Fund for Medical Science (CIFMS), China (grant number: 2018-I2M-2-002) and was supported as a Wellcome Trust Senior Investigator (grant 095541/A/11/Z). The Oxford Biobank is supported by the NIHR Oxford Biomedical Research Centre. The ISARIC4C study is supported by the Medical Research Council (grant MC_PC_19059), the National Institute for Health Research (NIHR) [award CO-CIN-01], the Medical Research Council [grant MC_PC_19059], the NIHR Health Protection Research Unit (HPRU) in Emerging and Zoonotic Infections at University of Liverpool [award 200907], NIHR HPRU in Respiratory Infections at Imperial College London [award 200927], and Liverpool Experimental Cancer Medicine Centre (Grant Reference: C18616/A25153), NIHR Biomedical Research Centre at Imperial College London [IS-BRC-1215-20013], and NIHR Clinical Research Network. J.C.K. is a Wellcome Investigator (WT204969/Z/16/Z) and supported by NIHR Oxford Biomedical Research Centre and CIFMS. LT is supported by a Wellcome fellowship (205228/Z/16/Z). E.J. is employed by INSERM, and the expenses for his work, which was performed at the IPBS, Toulouse, France, were funded by a private donation. The Spike Glycoprotein RBD from SARS-Related Coronavirus 2 Wuhan-Hu-1 (for Fig. 6), Recombinant from HEK293 Cells, NR-52306 was produced under HHSN272201400008C and obtained through BEI Resources, NIAID, NIH. The authors would like to thank the healthy volunteers who kindly donated their O−ve red cells for titration of samples, Jacques Izopet and Laurence Abravanel (Toulouse) for providing a set of positive and negative serum samples, and Olivier Bertrand for providing us with the corrected version of the IH4 amino acid sequence from the patent number US 9,879,090 B2. The views expressed are those of the author(s) and not necessarily those of the NHS, the NIHR, the Department of Health, or Public Health England.

## Author contributions

Conceived, initiated, and followed the project, documented portability and robustness of HAT method in a separate laboratory, FACS analysis, recruited collaborators, and handled manuscript submission: E.J. Designed and produced the IH4-RBD reagent, established conditions for lyophilisation, isolated and expanded human monoclonal antibodies to the RBD and ACE2-Fc, performed the standard HAT assays, and wrote the paper: A.T., P.R., J.X., T.K.T., L.S., J.H., R.R., and K.-Y.A.H. Contributed examples of HAT as a point of care test: N.G. Provision of serum/plasma sample sets for Fig. 4: Project management for Oxford serology work for sensitivity and specificity measurement and assessment of preliminary data: P.C.M., D.C., S.H., N.S.; collection and processing of samples, coordination of sample banks and running Siemens assay, ethics, storage of pre pandemic samples, coordination of provision of pre pandemic samples from Oxford BioBank, and seropositive donors through NHSBT: T.S., J.R., F.K., M.N., R.P., D.J.R., A.B., R.V., M.O., A.A.L., and H.P.T. Provision of Serum sample sets for Fig. 5 and Table 1 (COMBAT samples): A.J.M., J.C.K., A.J.K., P.K., and C.D.; ISARIC4C investigators: K.B., S.C.M., P.J.M.O., M.G.S., and L.C.W.T.; Oxford Immunology Network Covid-19 Response Clinical Sample Collection Consortium: M.A., A.A., S.B., S.B., E.C., A.E., M.M., D.G., T.L., J.M., E.P., V.S., G.S., A.S., and H.T.; OPTIC Clinical team: S. D., E.B., D.S., L.S., K.J., D.O'D., C.P.C., J.F., and C.V.A.-C.

## Competing interests

The authors declare no competing interests.

## Additional information

[1]MRC Human Immunology Unit, MRC Weatherall Institute, John Radcliffe Hospital, Oxford, UK. [2]Chinese Academy of Medical Science (CAMS) Oxford Institute (COI), University of Oxford, Oxford, UK. [3]Research Center for Emerging Viral Infections, College of Medicine, Chang Gung University, Taoyuan, Taiwan. [4]Division of Pediatric Infectious Diseases, Department of Pediatrics, Chang Gung Memorial Hospital, Taoyuan, Taiwan. [5]Structural Biology, The Rosalind Franklin Institute, Didcot, UK. [6]Vaccine Immunology Laboratory, National Institute of Immunology, New Delhi, India. [7]Department of Biochemistry, University of Oxford, Oxford, UK. [8]Department of Microbiology and Infectious Diseases, John Radcliffe Hospital, Oxford, UK. [9]Nuffield Department of Medicine, University of Oxford, Oxford, UK. [10]Nuffield Department of Medicine, University of Oxford, John Radcliffe Hospital, Oxford, UK. [11]Oxford NIHR Biomedical Research Centre, John Radcliffe Hospital, Oxford, UK. [12]Oxford University Hospitals NHS Foundation Trust, Oxford, UK. [13]Nuffield Department of Population Health, Big Data Institute, University of Oxford, Oxford, USA. [14]Oxford Centre for Diabetes, Endocrinology and Metabolism, Radcliffe Department of Medicine, University of Oxford, Oxford, UK. [15]Nuffield Department of Surgical Sciences, University of Oxford, John Radcliffe Hospital, Oxford, UK. [16]NHS Blood and Transplant, John Radcliffe Hospital, Oxford, UK. [17]BRC Haematology Theme and Radcliffe Department of Medicine, University of Oxford, John Radcliffe Hospital, Oxford, USA. [18]Public Health England, Salisbury, UK. [19]Wellcome Centre for Human Genetics, University of Oxford, Oxford, UK. [20]Division of Medical Sciences, University of Oxford, John Radcliffe Hospital, Oxford, UK. [21]Dengue Hemorrhagic Fever Research Unit, Office for Research and Development, Faculty of Medicine, Siriraj Hospital, Mahidol University, Bangkok, Thailand. [22]Oxford Vaccine Group, Department of Paediatrics, University of Oxford, Oxford, UK. [23]NIHR Oxford Biomedical Research Centre, Centre for Clinical Vaccinology and Tropical Medicine, University of Oxford,

Oxford, UK. [24]Genetics and Genomics, Roslin Institute, University of Edinburgh, Edinburgh, UK. [25]NIHR Health Protection Research Unit in Emerging and Zoonotic Infections, Institute of Infection, Veterinary and Ecological Sciences, University of Liverpool, Liverpool, UK. [26]National Heart and Lung Institute, Faculty of Medicine, Imperial College London, London, UK. [27]Institute of Pharmacology and Structural Biology (IPBS), University of Toulouse, CNRS, Toulouse, France. ✉email: alain.townsend@imm.ox.ac.uk; atnjoly@mac.com

