## [Peer Review File · Nature Communications]

Reviewers' Comments:

Reviewer #1:

Remarks to the Author:

I enjoyed reviewing the manuscript by Townsend et al, describing a Haemagglutination Test (HAT) for the detection of antibodies to SARS-CoV-2. The team of authors made a remarkable effort to produce and offer a low cost, easy to deploy test in resource limited settings for the detection of antibodies to SARS-CoV-2. This is an important public health need. They mainly explore the performance of the test in the context of seroprevalence studies.

The choice of RBD as the antigen for the HAT is fully explained and justified as it is expected to better correlate with neutralising antibodies and potentially the risk of re-infection/level of protection (the authors correctly point out this is a hypothesis for now).

In 148-233 the authors describe the development of the IH4-RBD reagent used downstream in the HAT, the optimisation of the concentration proposed and its specificity. The team completed experiments investigating 15 antibodies known to bind the RBD ; 12/15 produced an agglutination result. In the 3 that failed to agglutinate RBCs this was achieved with the addition of a mAb to human IgG.

Below my summary notes of the clinical evaluation against a CE marked RBD ELISA that was undertaken; to aid the review. (Table sent by email)

Comment 1: It is not clear in the manuscript if the addition of anti-IgG (described in the introduction as improving sensitivity) is recommended and/or was followed and what the potential impact on specificity is.

Comment 2: The choice of plasma over serum; there needs to be a sentence on why plasma was chosen and whether further evaluation for sera will be required. A relevant reference <https://journals.plos.org/plosone/article?id=10.1371/journal.pone.0048229>

Comment 3: Plasma of patients with other acute viral illness ie respiratory viruses like influenza/importantly, seasonal CoV/RSV and plasma from acute viral infections with the presence of IgM (ie CMV, EBV etc) were not tested.

Importantly for a HAT test plasma/sera from populations such as pregnant women, patients with autoimmune diseases where the presence of IgM might nbe an issue should be included.

Comment 4: Precision was not formally assessed; ie repeatability (intra-assay) and reproducibility (inter-assay variability) are not reported apart from the two operators reporting on photographs of plate results Running a sample multiple times and a panel by different operators (not only read) should be done and CV values reported. Is there potentially for digital read-out? Same for measurement of uncertainty ie testing of controls.

Comment 5: A summary of similar tests in development ie lateral flow and where the HAT test stands in terms of potential advantages and disadvantages would be useful.

Comment 6: both online links are not working and need updating

Comment 7: Line 353 onwards : the authors refer to rising titres post admission rather than post onset of symptoms ; this can be misleading. In addition, were these cases of hyperinflammation? We assume the critical were? It would help to clarify

Comment 8: the claim for use as POCT with capillary samples is limited as acknowledged and an

appropriate field study is hopefully underway where the stability of the reagents should also be properly tested

Comment 9: cost should include all related costs

Comment 10: Reading of results ; given the subjectivity of this in HAT an effort should be made to move on to a digital solution with the potential of utilising AI algorithms for reading results in a more reproducible way.

Reviewer #2:

Remarks to the Author:

Townsend, et al. presents a very innovative study developing a cheap, fast test toward identifying SARS-CoV-2 antibodies. Given the simplicity of the test and low complexity, there is potential for it to be significantly scaled toward a variety of health care settings including low-resource countries. The study effectively builds on previous investigations studying rapid hemagglutination tests using fusion proteins to detect HIV antibodies. The work is impressive for its thoroughness and detail in investigating a large number of COVID-19 samples, different monoclonal antibodies to the receptor binding domain (RBD) and testing across two different continents. The thoroughness of the investigation significantly increases the impact of the study. The willingness of the investigators to supply reagent to the scientific community is also applauded. The ability of the assay to titrate antibodies is also impressive, as well as the comparison to a clinical ELISA results for SARS-CoV-2 antibodies. A very detailed presentation of the methods is also commended.

Comments for the authors:

1. While entirely necessary, the authors could provide additional data showing the IH4-RBD fusion protein binds to the red blood cell membrane by staining the RBC with IH4-RBD with a FITC-labeled RBD antibody, an imaging under fluorescent microscopy and or flow cytometry. This would illustrate a bit more of the mechanism of the assay. (see Shao, Cellular & Molecular Immunology 2008, Figure 8).
2. It would be interesting to test if a His-tag antibody could also be used as a positive control reagent for the assay considering a His-tag antibodies might be more widely available.
3. The authors should add an explanation and discussion on the development of the assay, specifically the time point of one hour for the agglutination test. Previous studies in this field for HIV antibody identification also used a similar 96 well plate format (Shao, Cellular & Molecular Immunology 2008; Gupta, J Clin Microbiol. 2003), but the results were ascertained within several minutes, while the current study takes one hour.

While there is certainly utility in the test even for one hour , it would be interesting to present in a supplemental figure the status of the agglutination at earlier time points over one hour , and or if the assay could be optimized to act quicker. In addition to exploring this the authors could maybe comment on why the assay was unable to work in a faster format.

4. Another idea to explore is if the gravity process for the button to form in one hour is too slow, if the investigators could briefly centrifuge the plate to cause agglutination, followed by testing the assay by the same visual agglutination rules. This approach would resemble how tube testing for blood typing occurs in blood banks around the world today. Clearly, the disadvantage is the requirement for a machine, but for the sake of further investigating the assay, it would be interesting to see if the results could be ascertained in a faster way with this approach. As it stands, the one hour limitation would practically keep the say in a more centralized medical facility

and not for in the field testing applications how like lateral flow assays, this limitation could be mentioned in the discussion.

5. In the original Habib, et al. study, they were able to demonstrate agglutination of red blood cells in a slide agglutination format (see Figure 5). Have the investigators attempted a similar experiment to the original Habib study, since this slide agglutination format is or common in blood bank facilities?

6. I am curious if the assay would still work on a whole blood sample from a patient recovered from SARS-CoV-2, after the addition of the protein reagent this would require less sample prep and dilution (i.e. no dilution of RBC or plasma) than explored in the current assay. In figure 6, this was partially explored, although dilution was still used. The investigators could pursue this either in a 96 well-plate assay or slide assay for this purpose, but this would be one step closer toward clinical translation, consistent with other versions of this test on whole blood (Gupta, J Clin Microbiol. 2003).

7. The findings about the anti-RBD monoclonal antibodies that bound but did not agglutinate the red blood cells are very interesting. I wonder in a final version of this assay if anti-human IgG (AHG) should always be added to do the agglutination reaction or if it would cause too many false positives. Do the authors think this might enhance the sensitivity of the assay? it might be worth pursuing with a small number of samples.

8. I would add a point in the discussion section that controls would be needed in the clinical application since due to a low but significant prevalence of cold agglutinins that would cause apparent agglutination without requiring any SARS-CoV-2 antibodies. Furthermore, always using allogeneic RBC's as in this study would also lend the chance of a false positive from RBC alloantibodies.

9. The level of disagreements between the two blinded observers seems rather high (7%). The issue would be that the clinical use of the test would likely only have one observer leading to worse than the sensitivity and specificity reported. I realize the test is early in development, so one may just add a comment on this as a limitation in the discussion section. The subjectivity of the assay is certainly a challenge for its clinical implementation versus other serology tests.

10. There has been a lot of discussion in the laboratory testing community that these cheaper tests albeit having lower sensitivity, are nonetheless as useful as or high powered assays on clinical machines, if the same test is repeated twice. I'm curious if these samples that were "missed" could be repeated again and perhaps show a different result improving the performance of the assay.

11. This is a small point, but in the discussion section it might be useful to add some performance numbers on lateral flow at devices to give the reader understanding of how the hemagglutination based test is better. There is a reference to this, but would appreciate seeing the comparison side by side.

Reviewer #3:

Remarks to the Author:

Townsend et al. present a well-written description of their novel haemagglutination test (HAT), which evaluates the presence of antibodies directed against the receptor binding domain (RBD) of the spike protein from SARS-CoV-2. The presence of antibodies that bind SARS-CoV-2's RBD provide useful biomarker of immune response due to SARS-CoV-2 infection (e.g., evaluating infection or evaluating seroconversion after vaccination). Specifically, the authors describe a red blood cell HAT approach that makes use of an anti-glycophorin A (red cell protein) VHH antibody

domain fused to the COVID-19 RBD. This antibody reagent provides an intermolecular linkage for agglutination where a visible readout occurs in the presence of anti-RBD antibodies in the test sample. The read-out of the assay can be easily detected using nothing more than red blood cells, a "V-bottom" well microtiter plate, a trained reader, and time (after dilutions, approximately 60 minutes). Overall, the assay does quite well in their testing. They illustrate the HAT technique correlates well with existing commercial tests to detect antibodies targeting the RBD, and importantly, the antibody fusion reagent appears capable of tolerating lyophilization for ease of distribution. Furthermore, there is significant cost advantage to the HAT approach, as the large-scale production of the antibody fusion reagent (1-gram scale) is enough for 10 million tests (at ~0.27 UK pence/ < \$0.01 USD). The authors provide limited but encouraging direct point of care tests on capillary blood samples, which provides a roadmap of even more potential for this HAT strategy. Finally, the authors will ship lyophilized aliquots (good for ~10,000 tests, free of charge).

The limitations of their HAT approach, particularly the sensitivity (~90%) and specificity (99%), which do not fall completely above recommended government standards (98% - 99.5 % for both sensitivity and specificity), are clearly acknowledged, while providing a compelling argument for the use of HAT despite falling under these guidposts.

This work will certainly be of interest to the community and broader field alike. It is very likely to influence and shape low cost alternatives to develop serological COVID-19 tests and beyond, especially in low and middle-income countries.

Minor comments:

- The authors mention a group that has described preliminary results using an approach similar to that described here, except a scFV antibody against a different red blood cell H antigen was used. Might the authors provide some discussion on potential advantages of VHH domain vs. scFV (e.g., potential costs, storage, etc.)
- Figure 2B: It is not clear if samples are duplicates.
- Page 7, line 177: The authors' state: " (Figure 2A Row 8)," why not refer to this the bottom, "Nil" row.

Reviewer #4:

Remarks to the Author:

A haemagglutination test for rapid detection of antibodies 1 to SARS-CoV-2

Alain Townsend^{1,2*}, Pramila Rijal^{1,2}, Julie Xiao¹, Tiong Kit Tan¹, Kuan-Ying A Huang³, Lisa Schimanski^{1,2}, Jiangdong Huo⁴, Nimesh Gupta⁵, Rolle Rahikainen⁶, Philippa C Matthews^{7,8}, Derrick Crook^{7,9,10}, Sarah Hoosdally^{8,10}, Teresa Street⁹, Justine Rudkin¹¹, Nicole Stoesser^{7,9}, Fredrik Karpe¹², Matthew Neville¹², Rutger Ploeg¹³, Marta Oliveira¹³, David J Roberts^{14,15}, Abigail A Lamikanra¹⁴, Hoi Pat Tsang¹⁴, Abbie Bown¹⁶, Richard Vipond¹⁶, Alexander JMentzer¹⁷, Julian C Knight¹⁷, Andrew Kwok¹⁷, Gavin Screatton^{17,18}, JuthathipMongkolsapaya^{2,18,19}, Wanwisa Dejnirattisai¹⁷, Piyada Supasa¹⁷, Paul Klenerman⁸, ChristinaDold^{20,21}, Kenneth Baillie²², Shona C Moore²³, Peter JM Openshaw²⁴, Malcolm G Semple²³, Lance CW Turtle²⁵, Mark Ainsworth²⁶, Alice Allcock¹⁷, Sally Beer²⁶, Sagida Bibi²⁰, ElizabethClutterbuck²⁰, Alexis Espinosa²⁶, Maria Mendoza²⁶, Dominique Georgiou²⁶, Teresa Lockett²⁶, Jose Martinez²⁶, Elena Perez²⁶, Veronica Sanchez²⁶, Giuseppe Scozzafava²⁶, AlbertoSobrinodiaz²⁶, Hannah Thraves²⁶, Etienne Joly^{27*}

Townsend and colleagues present a haemagglutination assay for use in the detection of SARS-CoV-2 antibodies in serum/plasma. The assay consists of a nanobody-antigen fusion protein that effectively serves to cross-link or agglutinate red blood cells (RBCs) to one another in the presences of antibodies targeting the antigen of interest. In their assay system, the nanobody, VHH-1H4, targets Glycophorin A, a protein that is highly expressed on the surface of RBCs. The

nanobody is C-terminally fused to the receptor binding domain of the SARS-CoV-2 spike protein. Since antibodies are bivalent (i.e., IgG, IgA) or pentameric (IgM), one molecule is able to engage multiple RBC and high-order complexes are formed.

In this hemagglutination assay, they show that when RBCs from a naïve donor is mixed with their fusion protein and serum spiked with monoclonal antibodies or serum from COVID-19 subjects, the RBCs agglutinate in an anti-RBD antibody concentration dependent manner. The data in the manuscript shows that the test functions with monoclonal antibodies, convalescent serum and COVID-19 hospitalized patient serum obtained within the first 5 days of hospitalization. They also performed testing on serum obtained from patients from the pre-COVID-19 era to assay for the tests specificity. The authors report a sensitivity of ~90% and a specificity of ~99% in convalescent patients. The assay is relatively cheap ~£26 per plate. The benefit of this assay is that the assay requires little in the way of equipment, cold chain and molecular biology skills. However, I have significant concerns about the significance of this assay given its modest performance characteristics, other faster tests with similar performance (e.g., Binax Now) and the assay being incomplete (i.e., missing controls in the working setup and the lack of formal validation testing). These latter aspect is especially important as the authors mention multiple times how they are willing to provide materials to others, suggesting that the assay is ready for large scale use and it is not.

Specifically concerns include:

- 1) There is no data on RBC donor-to-donor variations or RBC age impacts the assay sensitivity and specificity.
- 2) Other than testing a set of pre-covid-19 serum samples, there is no formal evaluation on the impact of RBD targeting antibodies against endemic coronaviruses on the assay. Since the negative serum were not tittered against endemic coronaviruses, the specificity number may not be accurate.
 - a. Similarly, ideally more than one antigen would be used targeting epitopes that are specific to SARS-CoV-2. This would be expected to increase sensitivity and perhaps specificity.
- 3) While I applaud that the assay controls for the presence of anti-RBC targeting antibodies, there is a striking lack of a negative and positive control serum. Well characterized reagents for this purpose should be included in the assay on every plate. This can be a home-made or commercially provide reagent. If end-users are going to be expected to provide the material for their assay, the ideal performance characteristics of the controls should be well described, to allow for end-users to perform the assay appropriately.
- 4) Early sensitivity is being defined as the first five days of hospitalization, but the data should be discussed in time post-symptom onset as the time to presentation is variable and has the potential to bias the sensitivity higher or lower.
- 5) Most critically, like other serologic assay's its highest sensitivity is in convalescent patients and is not surprisingly least sensitive during early illness. This limit's its use to population studies and for post-facto diagnosis of people who were not tested early on and is therefore, unlikely to have a significant impact on lowering disease transmission dynamics.

POINT BY POINT REPLY TO REVIEWER COMMENTS

Reviewer #1 (Remarks to the Author):

I enjoyed reviewing the manuscript by Townsend et al, describing a Haemagglutination Test (HAT) for the detection of antibodies to SARS-CoV-2. The team of authors made a remarkable effort to produce and offer a low cost, easy to deploy test in resource limited settings for the detection of antibodies to SARS-CoV-2. This is an important public health need. They mainly explore the performance of the test in the context of seroprevalence studies.

The choice of RBD as the antigen for the HAT is fully explained and justified as it is expected to better correlate with neutralising antibodies and potentially the risk of re-infection/level of protection (the authors correctly point out this is a hypothesis for now).

In 148-233 the authors describe the development of the IH4-RBD reagent used downstream in the HAT, the optimisation of the concentration proposed and its specificity. The team completed experiments investigating 15 antibodies known to bind the RBD; 12/15 produced an agglutination result. In the 3 that failed to agglutinate RBCs this was achieved with the addition of a MAb to human IgG.

Below my summary notes of the clinical evaluation against a CE marked RBD ELISA that was undertaken; to aid the review. (Table sent by email)

Comment 1: It is not clear in the manuscript if the addition of anti-IgG (described in the introduction as improving sensitivity) is recommended and/or was followed and what the potential impact on specificity is.

Response:

The reviewer is actually mistaken – we do not mention the addition of anti-IgG anywhere in the introduction or ever state that it increases the sensitivity of the test, because we did not attempt to measure this for several reasons enumerated below. We made absolutely clear in the methods section that anti-IgG was not involved in the standard HA test on serum or plasma samples. We have distributed the test to > 20 interested groups around the world, and not one of them found this confusing or has queried the addition of an anti-IgG reagent. We used anti-IgG simply as an indirect Coombs test to investigate whether the monoclonal antibodies that failed to agglutinate the IH4-RBD coated red cells, had bound without agglutinating.

It is possible that addition of anti-IgG would increase the overall sensitivity of the test, in conditions where low titre antibodies had bound the red cells but failed to agglutinate them. This is, however, a very complex issue. As the reviewer must know interventions that increase the sensitivity of a test very frequently degrade its specificity. For this test the specificity is the most important measure – a false positive result is potentially more dangerous than failing to detect low titre antibodies. To find out if addition of an anti-IgG preparation increased sensitivity we would have to repeat all of the experiments in the paper, and re-measure the sensitivity and specificity in the two settings we studied in detail. Furthermore, the addition of an anti-Ig reagent would add considerably to the cost of the test. As we discuss in the paper we have deliberately kept the methods as simple as possible, and offer the test as it is with Sensitivity ~90% and Specificity at least 99%.

Comment 2: The choice of plasma over serum; there needs to be a sentence on why plasma was chosen and whether further evaluation for sera will be required. A relevant reference <https://journals.plos.org/plosone/article?id=10.1371/journal.pone.0048229>

Response:

We did not choose plasma over serum. We tested all of the samples we managed to obtain, whether they were serum or plasma. We have not detected any obvious difference between the two. Again, to really look for a difference in sensitivity and specificity in clinical samples from sero-positive patients is not at all simple. We would have to re-collect paired samples of serum and plasma from all of the positive and negative donors and compare the titres. This is simply not a practical possibility.

Action:

To gain insight into this we have obtained from sero-negative donors both fresh serum as well as plasma from K2ETDA and Heparin samples. We then spiked these with a selection of monoclonal Class 1-4 Monoclonal antibodies (to each of the four antibody binding sites defined by Barnes et al 2020), and compared the titres obtained. We did this in two ways to mimic the conditions in figure 4 (where 1:40 serum/plasma was maintained throughout the titration) and figure 5 (where the doubling dilutions were done in PBS).

Result:

As can be seen on slides 1-13 of the pdf provided, in both situations the end-points were the same – the presence of serum had an effect on the morphology of the agglutinated red cell pellet, but no effect on the agglutination end-points. Therefore, we are confident that in neither the screening assay at 1:40 dilution (Fig 4), nor in the titration assay (Fig 5) did the serum or plasma origin of the samples alter the sensitivity of detection of antibodies to the RBD.

Manuscript:

A paragraph describing these results has been inserted, Lines 238-246

Comment 3: Plasma of patients with other acute viral illness i.e. respiratory viruses like influenza/importantly, seasonal CoV/RSV and plasma from acute viral infections with the presence of IgM (i.e. CMV, EBV etc) were not tested.

Response:

We tested 54 samples from patients with sepsis, all of which tested negative, and a total of 224 samples from other control donors giving a total of 278 negative samples with a single weak false positive result in the assays to assess sensitivity and specificity. Many of these donors are expected to have had high titre antibodies to influenza and other common infections, including common cold coronaviruses. Since those were clearly not detected in the test, we do feel that we have shown a reasonable data set to establish that, if there is some cross reactivity to other viruses, it is a very infrequent problem for HAT, and that the test is at least 99% specific. To do as the referee asks would require a properly designed prospective study to collect blood samples from a large cohort of patients with respiratory illness for whom gold standard PCR diagnosis of each infection was available. This is simply not possible for this paper.

As a further elaboration on this issue, we know from published data that ~90% of adults are seropositive for OC43 (the most closely related of the common cold Betacoronaviruses), (Severance et al; <https://www.ncbi.nlm.nih.gov/pmc/articles/PMC2593164/>; and Anderson et al BioRxiv 2020, <https://www.medrxiv.org/content/10.1101/2020.11.06.20227215v1>), and yet we are not detecting a high false positive rate in the pre-covid samples. This is because the great majority of this cross-reactivity is directed at the N protein (Severance et al) and the S2 domain of the Spike protein (Ng et al, [https://science.sciencemag.org/content/early/2020/11/05/science.abe1107](https://science.sciencemag.org/content/early/2020/11/05/science.abe1107;);)). In the latter paper all of the cross-reactivity to the spike protein was absorbed out with the isolated S2 domain, but NOT by the S1 domain containing the RBD. This is very likely because there is very much less conservation of the RBD sequence between these viruses (~22% v 43% for the S2 domain). In our own work characterising monoclonal antibodies to SARS-CoV-2 (Huang et al, <https://www.biorxiv.org/content/10.1101/2020.08.28.267526v1>;) we found 5 out of 9 MAbs to S2 contained sequence features of memory cells (high level of somatic mutation) and cross-reacted on OC43 S2, and a similar proportion for N. For the RBD we found just 1 of 31 MAbs that cross-reacted with RBD from OC43. This single antibody was close to germline in sequence (i.e. Not derived from a memory B cell), and in any case failed to cross-link red cells in the HA test.

Finally, in our two assays to measure the operating characteristics of the HA test, the single false positive for each run was derived from a single well that showed a partial teardrop in one assay (scored negative) and loss of teardrop in the second (scored positive). In each case we investigated these by repeat measurements, and a titration with a reagent composed of IH4 without the RBD. In both cases the repeat assays showed only a partial teardrop in the first well of the titration, showing that whatever antibody was causing the reaction was low titre. In one case this weak reaction was clearly dependent on the presence of the RBD, i.e. could have been due to rare and very low titre cross-reactive antibodies between the RBD of a common cold virus and SARS-CoV-2. This is consistent with detection in 0.6-2% of pre-covid donors by Anderson et al. (see above) of binding activity to the RBD of SARS-CoV-2. In the second case the reaction was with the IH4 component. Lastly, we have heard from our colleague Prof Gathsaurie Malavige in Sri Lanka, who has been using the HA test successfully for some time, and found a single false positive in 110 pre-covid samples, from a patient with Dengue.

So, with this evidence in hand, we feel that false positive results in the HA test due to very low-level cross-reactivity between the RBDs of common cold betacoronaviruses and the RBD of SARS-CoV-2 will be rare (less than 1%), and can be distinguished by their low titre.

Comment 3b: Importantly for a HAT test plasma/sera from populations such as pregnant women, patients with autoimmune diseases where the presence of IgM might be an issue should be included.

Response:

We do not follow the referee's reasoning here. If by "autoimmune diseases where the presence of IgM might be an issue" the referee means diseases in which there may be autoantibodies to red cells (such as cold agglutinins) that might cause false positive results – we controlled for that by always having a parallel well with PBS added instead of the IH4-RBD reagent in the screening test at 1:40 dilution (Fig 4). None of the 199 pre-covid samples gave a positive result without the IH4-RBD reagent. If the serum contained antibodies that

spontaneously agglutinated the red cells we would have detected them, and in the methods section, with the protocols listed, we stipulate that in the screening test at 1:40 of serum/plasma this PBS control should be always included. Again, we do not follow the referee's concern about "pregnant women". If these donors had autoagglutinating antibodies they would be detected in the parallel control well containing just PBS. While we would like to expand the range of PCR negative donors to provide more information on these issues, we feel that, as things stand, we have done enough to provide sufficient information for the test to be used correctly by trained practitioners.

Comment 4: Precision was not formally assessed; i.e. repeatability (intra-assay) and reproducibility (inter-assay variability) are not reported apart from the two operators reporting on photographs of plate results Running a sample multiple times and a panel by different operators (not only read) should be done and CV values reported. Is there potentially for digital read-out? Same for measurement of uncertainty i.e. testing of controls.

Response:

*We are familiar with a slightly different formal vocabulary for Precision: Repeatability = within assay variation; Intermediate Precision = inter assay variation within a laboratory; Reproducibility = variation in test results between different laboratories. We showed clearly that the results between assays in our lab repeated on the 297 test samples used to calculate sensitivity and specificity were close (Figure 4). This was the case also on the tests done on donors in the first 5 days of admission, where the repeat on 232 samples was performed 34 days later using a different donor of the O-ve indicator red cells (figure 5 and S2). This shows good **Intermediate Precision** (within laboratory, different operators), and correlated very well with an independent gold standard (Siemens Chemiluminescence test). We have also included in supplementary Tables 3 and 4 data showing the reproducibility in our lab with O-ve red cells from different donors, where for a set of 4 Monoclonal antibodies, 4 Sera, and 3 O-ve donors the HAT titrations varied at most by +/- 1 doubling dilution. This shows that the test gives good **Repeatability**: in our numerous experiments, variations between replicates within an assay were never greater than one doubling dilution. Finally, in a collaboration with Rebecca Cox and Sonja Ljostveit in Bergen Norway, we have found true **Reproducibility** between laboratories with different operators where an examination of 220 samples (87% PCR positive) from a cohort of donors with community acquired infections showed the HAT gave 91% sensitivity and 100% specificity (gold standard PCR+ve or seropositive by ELISA for anti-Spike EPT). This latter study will be published elsewhere as part of a review of Norway's COVID-19 burden.*

Action:

To explore the issue of possible variations in HAT sensitivity due to individual differences in red cells' characteristics, we have compared HAT performance using O-ve Red cells from three different donors and used them to perform titrations of Class 1-4 Monoclonal Antibodies to the RBD, together with a MAb to Ebola as a negative control, and four selected sera with high (9DD = 1:10,240), middle (6DD = 1:1,280), low (3DD = 1:160), and negative titres, all performed in quadruplicates.

Result:

Similarly to all the experiments shown in the paper, the results of those experiments, which can be seen on slides 14-16 of the pdf file provided, show maximum variation between all of the measurements of +/- 1 doubling dilution, suggesting that individual variations of red cells characteristics have very little influence on the performances of HAT.

Manuscript:

We have included this data in Supplementary tables 3 and 4 and described the results on lines 278-284 of the manuscript.

Regarding the possibility of a digital read-out, see response to comment 10.

Comment 5: A summary of similar tests in development i.e. lateral flow and where the HAT test stands in terms of potential advantages and disadvantages would be useful.

Response:

Results on this type of test are accumulating rapidly. We had referred the reader to a review of measurements (Adams et al from the UK National COVID Testing Scientific Advisory Panel) and feel that this is sufficient.

Comment 6: both online links are not working and need updating

Response:

We have fixed this.

Comment 7: Line 353 onwards : the authors refer to rising titres post admission rather than post onset of symptoms ; this can be misleading. In addition, were these cases of hyperinflammation? We assume the critical were? It would help to clarify.

Response:

We have clearly stated that 1) the symptoms are likely to have started several days before admission and do not feel that this statement is misleading. 2) made clear that these samples are from a range of patients including critical and severe (as defined in methods) who had severe inflammation. A full description of these patients will be published from Oxford collaborators in due course.

Comment 8: the claim for use as POCT with capillary samples is limited as acknowledged and an appropriate field study is hopefully underway where the stability of the reagents should also be properly tested.

Response:

Somewhat ironically, although the capacity to use capillary blood should considerably simplify things for the test to be performed as a POCT, because whole blood cannot be kept for more than two or three days, it makes things much less straightforward than using serum or plasma to carry out the study necessary to optimize and validate the test. A study to address all these issues is indeed under way. But this type of study will take much more time to complete because, since it entails using whole blood which has to be obtained from patients rather than from collections held in freezers, it requires addressing many ethical issues, and going through lengthy administrative steps.

Comment 9: cost should include all related costs.

Response:

We do not feel that this is a reasonable request because the costs will vary depending on the settings, and the country, if only because the time of the operator(s) will be the most costly item in most cases. Other costs will be for blood-drawing equipment, plastic plates (which could, if needed, be washed and used again), re-usable adjustable pipettes, pipet tips and PBS-based buffers. The costs of these will never be very high, but will vary by factors of several fold depending on quantities, as well as between countries.

*The key point is that the cost of the single IH4-RBD reagent is **0.27 pence per test well** . And this cost will be covered by us, the provider, and nothing to those who wish to use it, as we are providing enough for 10 million test wells through donations, and will provide more if requested.*

Comment 10: Reading of results ; given the subjectivity of this in HAT an effort should be made to move on to a digital solution with the potential of utilising AI algorithms for reading results in a more reproducible way.

Response:

We have looked into this. It would be expensive and time consuming to establish and distribute. The whole point of this test is that it should be available to healthcare workers providing a service in difficult underfunded conditions, and in remote locations, where it can be performed by members of staff in a side room on the ward, or in the community in family settings. The great strength of this assay is that it can be read by eye without any sophisticated machines or algorithms. We have spent much time in finding a method to provide an objective end-point clearly visible by eye , and feel that the criterion we have settled for, i.e. the loss of teardrop, matches those requirements very well. Other methods (similar to blood typing) really required a microscope, or at least a magnifying glass, to observe the agglutination, did not give clear objective endpoints, and were not adaptable to doing hundreds of titrations in a day.

Reviewer #2 (Remarks to the Author):

Townsend, et al. presents a very innovative study developing a cheap, fast test toward identifying SARS-CoV-2 antibodies. Given the simplicity of the test and low complexity, there is potential for it to be significantly scaled toward a variety of health care settings including low-resource countries. The study effectively builds on previous investigations studying rapid hemagglutination tests using fusion proteins to detect HIV antibodies. The work is impressive for its thoroughness and detail in investigating a large number of COVID-19 samples, different monoclonal antibodies to the receptor binding domain (RBD) and testing across two different continents. The thoroughness of the investigation significantly increases the impact of the study. The willingness of the investigators to supply reagent to the scientific community is also applauded. The ability of the assay to titrate antibodies is also impressive, as well as the comparison to a clinical ELISA results for SARS-CoV-2 antibodies. A very detailed presentation of the methods is also commended.

Comments for the authors:

1. While not (?) entirely necessary, the authors could provide additional data showing the IH4-RBD fusion protein binds to the red blood cell membrane by staining the RBC with IH4-RBD with a FITC-labelled RBD antibody, an imaging under fluorescent microscopy and or flow cytometry. This would illustrate a bit more of the mechanism of the assay. (see Shao, Cellular & Molecular Immunology 2008, Figure 8).

Response:

It is actually difficult to see how agglutination could work if the IH4-RBD reagent did not bind to the red cell surface, and we indeed make regular use of flow cytometry to obtain quantitative measurements of the anti-RBD antibodies bound to the surface of the red cells.

Action:

Following the referee's request, we have now included an additional panel to Figure 3, showing titrations of Monoclonal Antibody CR3022 and of a positive serum from a convalescent Covid-19 patient, read both by eye after HAT, and by flow cytometry after Indirect Immunofluorescence. This shows that for both the MAb and the serum, loss of teardrop occurs when bound antibody is well below saturation.

Manuscript:

We have included this result as Figure 3C and describe the results lines 271-276.

2. It would be interesting to test if a His-tag antibody could also be used as a positive control reagent for the assay considering a His-tag antibodies might be more widely available.

Response:

We have indeed used an anti His-tag monoclonal antibody (Invitrogen MA1-21315), and found that it does agglutinate the red cells in the presence of 100 ng/well of the IH4-RBD reagent. Following the referee's request, we have now included the information to supplementary table 1 that titration of this anti His-tag monoclonal antibody reached an end point at 62.5 ng/well. This is within the range for the end points for the monoclonal antibodies to the RBD. The latter are however made in house and we supply one with the IH4-RBD reagent for free as a control. The cost of the anti HIS reagent is exorbitant. Any group who establishes the assay will soon find donors with high titre antibodies that can provide an additional control, but we are quite prepared to re-supply the CR3022 MAb control as needed, or any of the other MABs we list in the paper.

3. The authors should add an explanation and discussion on the development of the assay, specifically the time point of one hour for the agglutination test. Previous studies in this field for HIV antibody identification also used a similar 96 well plate format (Shao, Cellular & Molecular Immunology 2008; Gupta, J Clin Microbiol. 2003), but the results were ascertained within several minutes, while the current study takes one hour.

While there is certainly utility in the test even for one hour, it would be interesting to present in a supplemental figure the status of the agglutination at earlier time points over one hour, and or if the assay could be optimized to act quicker. In addition to exploring this the authors could maybe comment on why the assay was unable to work in a faster format.

Response:

We are trained in haematology as well as in virology, but we found that the virological assay (very similar to titrating influenza virus, i.e. the loss of a teardrop in 96-well V-bottomed plates) simply worked better as a titratable assay than the flat-surface-based assay of haematologists. And the loss of the teardrop provided an objective end-point which can simply be read by eye. In the early days of the project, we did actually explore the possibility of using slide agglutination, and found that it does indeed work with our reagent. But whilst

it may indeed provide the means to get quicker results, we also found that it does not provide a clear end-point in titrations and is unwieldy for setting up hundreds of samples at a time.

*In order to keep our manuscript reasonably short, we chose not to dwell on the many attempts and steps we took to finally arrive at the test's protocol. As it stands, this protocol is actually the result of **considerable efforts** to keep the whole procedure as simple, reliable, reproducible and clear as possible.*

We have explained that the test depends on the red cells settling to the bottom of the V shaped test well to form a button. The agglutination is then detected by tilting the plate and showing that flow of the button down the wall of the well to form a "teardrop" is inhibited by the agglutination. The assay takes one hour because it takes one hour for the red cells to settle into a button at the bottom of the well.

In the paper by Shao & Zhang (2008) pointed to us by the referee, figure 9 indeed shows a picture of teardrops that are very similar to those in our essays, with red blood cells clearly collected at the bottom of the wells of the 'weak agglutination samples'.

Figure 9. Mimic autologous erythrocyte agglutination of bifunctional protein with 30 plasma samples from HIV-infected individuals. The arrows indicate the weak agglutination samples.

We must admit that we are slightly puzzled by the fact that those pictures could have been taken after only 5 minutes of incubation at room temperature because, in our hands, it takes at the very least 45 minutes for the red cells to sink to the bottom of the wells. Regarding the paper by Gupta et al., they did not use 96-well plates, but glass slides, and we have explained above why we chose not to use slide hemagglutination.

4. Another idea to explore is if the gravity process for the button to form in one hour is too slow, if the investigators could briefly centrifuge the plate to cause agglutination, followed by testing the assay by the same visual agglutination rules. This approach would resemble how tube testing for blood typing occurs in blood banks around the world today. Clearly, the disadvantage is the requirement for a machine, but for the sake of further investigating the assay, it would be interesting to see if the results could be ascertained in a faster way with this approach. As it stands, the one-hour limitation would practically keep the say in a more centralized medical facility and not for in the field-testing applications how like lateral flow assays, this limitation could be mentioned in the discussion.

Response:

We have actually also explored the possibility of using centrifugation to shorten the incubation time, but found that some blood samples tend to lose the formation of teardrops

more than others after centrifugation, and this introduces a variability which is well-nigh impossible to control.

Even though it takes one hour for the test to develop and a few minutes to be read, this is still a much shorter time to a result than provided in most western hospitals for any blood test. These usually take several hours for a result to be reported. In addition, although one test takes one hour, the assays can be staggered and batches read every ten minutes. So, there is no real advantage to tests that take a few minutes to develop over one that takes an hour.

Also, in our experience the length of 1 hour is not a problem. Assays can be run in parallel (by simply noting the time of set up on the plates) and read accurately in batches every few minutes. There is no need for a “centralised medical facility” – the whole point is that this assay can be set up and read in a side room on a ward in a remote hospital, or in the community.

*To conclude on this point of trying to reduce the incubation time, there are, no doubt, many possible variations on this theme that could be tried, and the time taken to develop the test could possibly be shortened. Each such variation would, however, have to be put through the full gamut of sensitivity and specificity assessment. If we change anything at all, **all of these tests would have to be repeated to ensure the changed method retained the same operating characteristics for sensitivity and specificity.** It would be inappropriate to start suggesting untried modifications, because if they were to be used, they could provide unreliable results.*

5. In the original Habib, et al. study, they were able to demonstrate agglutination of red blood cells in a slide agglutination format (see Figure 5). Have the investigators attempted a similar experiment to the original Habib study, since this slide agglutination format is or common in blood bank facilities?

Response:

As already stated above, the slide agglutination works. But we find that it does not provide a clear end-point in titrations and is unwieldy for setting up hundreds of samples at a time.

6. I am curious if the assay would still work on a whole blood sample from a patient recovered from SARS-CoV-2, after the addition of the protein reagent this would require less sample prep and dilution (i.e. no dilution of RBC or plasma) than explored in the current assay. In figure 6, this was partially explored, although dilution was still used. The investigators could pursue this either in a 96 well-plate assay or slide assay for this purpose, but this would be one step closer toward clinical translation, consistent with other versions of this test on whole blood (Gupta, J Clin Microbiol. 2003).

Response:

In the manuscript, we provide a clear protocol for a finger-prick version of the test on capillary blood in the paper. The referee is correct and the test can be done on the donors' own red cells, and this has obvious advantages. We have explained in the text that this type of test was foremost in our minds when we developed the HA test, and this is why we investigated using simple diluted whole O-ve blood for detection with serum/plasma samples, rather than washed red cells – to mimic the situation of capillary blood. Dr Nimesh Gupta (who, incidentally, is not the same as the author of the 2003 paper pointed to by the referee) volunteered to provide preliminary examples of the capillary blood test results from New

Delhi to show that the test worked in a distant laboratory using the reconstituted lyophilised reagent.

We are not sure what the referee means by “a dilution was used”. The capillary blood sample must, like O-ve indicator cells, be diluted 1:40 to provide the optimal number of red cells for the formation of a button at the bottom of the wells and subsequently of a teardrop.

We have of course performed the test on many capillary samples locally during the exploratory phase of developing the test. However, to do this properly, as we state in the paper, we would need to compare parallel serum/plasma samples from venous blood with capillary samples on a set of donors with a known prevalence and titre of positive serology. This would formally establish the operating characteristics of capillary sampling for this test.

As stated above, this is currently under way, but as we are sure the referee will appreciate, this is not simple either ethically or practically at the moment. It is relatively easy to acquire stored clinical serum/plasma samples as part of other studies – but the comparison to capillary sampling would require a venous and a finger-prick sample from each volunteer, immediate testing, and a level of co-exposure which involves some level of infection risk. Such studies are currently being planned, both in Oxford on volunteer Health Care Workers, and in Toulouse on left-over clinical blood samples. In these studies, we will compare a full titration of venous and capillary plasma with the donors own red cells and with O-ve red cells. For now, we have described how to do the test on capillary blood but acknowledge that we cannot, at the moment, quote operating characteristics for the test done this way.

7. The findings about the anti-RBD monoclonal antibodies that bound but did not agglutinate the red blood cells are very interesting. I wonder in a final version of this assay if anti-human IgG (AHG) should always be added to do the agglutination reaction or if it would cause too many false positives. Do the authors think this might enhance the sensitivity of the assay? it might be worth pursuing with a small number of samples.

Response:

*It is possible that addition of anti-IgG would increase the overall sensitivity of the test, in conditions where low titre antibodies had bound the red cells but failed to agglutinate them. This is, however, a very complex issue. As the reviewer suggests, interventions that increase the sensitivity of a test very frequently degrade its specificity. For this test the specificity is the most important measure – a false positive result is potentially more dangerous than failing to detect low titre antibodies. To find out if addition of an anti-IgG preparation increased sensitivity we would have to repeat all of the experiments in the paper, and re-measure the sensitivity and specificity in both of the two the settings we studied in detail (Figure 4 and 5). To show results on a few samples would be misleading, and might encourage users to use this modification without realising that they might be severely degrading the specificity of the test. In addition, the addition of an anti-Ig reagent would add considerably to the cost of the test. As we discuss in the paper we have deliberately kept the methods **as simple and clear as possible**, and offer the test as it is, with a single reagent that provides a Sensitivity ~90% and Specificity at least 99%.*

The Use of Anti IgG with the monoclonal antibodies that failed to agglutinate on their own was used as an indirect Coombs test to establish that the MAbs and nanobodies-Fc that failed to agglutinate had actually bound to the red cells. This situation is common in blood typing and it was Coombs’s breakthrough to increase the sensitivity of detection of pathology

causing “incomplete” anti-Rhesus antibodies by this means.

8. I would add a point in the discussion section that controls would be needed in the clinical application since due to a low but significant prevalence of cold agglutinins that would cause apparent agglutination without requiring any SARS-CoV-2 antibodies. Furthermore, always using allogeneic RBC's as in this study would also lend the chance of a false positive from RBC alloantibodies.

Response:

We refer the referee to the section in the paper where we discussed this very point (Line 325-327) and emphasised the inclusion of parallel wells containing serum, but PBS instead of IH4-RBD, to detect such antibodies (see figure 4, where every well was controlled in this way). We did not detect any such reaction in the 199 negative control samples. Hence, if any of the samples tested contained cold agglutinins, they did not agglutinate on their own under the conditions used for our test. But - as Coombs discovered - may well have done so if we had added an anti-IgG reagent.

9. The level of disagreements between the two blinded observers seems rather high (7%). The issue would be that the clinical use of the test would likely only have one observer leading to worse than the sensitivity and specificity reported. I realize the test is early in development, so one may just add a comment on this as a limitation in the discussion section. The subjectivity of the assay is certainly a challenge for its clinical implementation versus other serology tests.

Response:

We apologise – the “7%” figure for disagreements in the manuscript is a misprint – it should be 7/300 disagreements = 2.3%. Any test read by eye is bound to have an element of subjectivity. These disagreements occurred in the screening assay with 1:40 diluted samples, where the wells were positive in one assay and showed a partial teardrop in the other. That is to say where the settled button of red cells showed incomplete rather than full flow down the side of the well on tilting for 30 seconds. We have discussed this issue and recommended calling negative any wells with any partial flow, so as to provide the most objective end-point possible. If so desired these reactions could be further investigated with a titration and an alternative test such as ELISA if available.

10. There has been a lot of discussion in the laboratory testing community that these cheaper tests albeit having lower sensitivity, are nonetheless as useful as or high-powered assays on clinical machines, if the same test is repeated twice. I'm curious if these samples that were “missed” could be repeated again and perhaps show a different result improving the performance of the assay.

Response:

Please look at figure S2: for the 232 samples repeated after storage at 4° C for 34 days, on O-ve red cells from a different donor, 99% of the titrations were within one doubling dilution of each other, with correlation coefficient $R^2 = 0.94$. In addition, repeating the assay during the first 5 days in hospital is also informative - 67% increased in titre, but none reduced in titre. These results suggest that positive and negative results from stored samples remain consistent on repeat testing, and repeat sampling from

donors can “improve” the results in settings where the antibody level is likely to be rising with time.

11. This is a small point, but in the discussion section it might be useful to add some performance numbers on lateral flow at devices to give the reader understanding of how the hemagglutination based test is better. There is a reference to this, but would appreciate seeing the comparison side by side.

Response:

We would prefer not to do this – we had referred to a good current review by Adams et al– (UK National COVID Testing Scientific Advisory Panel) where the sensitivity values for these tests varied between 55%-70%, compared to 90% for the HAT in our assays. To do this comparison properly we would have to compare a lateral flow test alongside the HAT in all of the assays we describe. No doubt some of these lateral flow tests will eventually perform better than our HAT – but one thing we can guarantee – they will cost a very great deal more than 0.27 pence per test.

Reviewer #3 (Remarks to the Author):

Townsend et al. present a well-written description of their novel haemagglutination test (HAT), which evaluates the presence of antibodies directed against the receptor binding domain (RBD) of the spike protein from SARS-CoV-2. The presence of antibodies that bind SARS-CoV-2’s RBD provide useful biomarker of immune response due to SARS-CoV-2 infection (e.g., evaluating infection or evaluating seroconversion after vaccination). Specifically, the authors describe a red blood cell HAT approach that makes use of an anti-glycophorin A (red cell protein) VHH antibody domain fused to the COVID-19 RBD. This antibody reagent provides an intermolecular linkage for agglutination where a visible readout occurs in the presence of anti-RBD antibodies in the test sample. The read-out of the assay can be easily detected using nothing more than red blood cells, a “V-bottom“ well microtiter plate, a trained reader, and time (after dilutions, approximately 60 minutes). Overall, the assay does quite well in their testing. They illustrate the HAT technique correlates well with existing commercial tests to detect antibodies targeting the RBD, and importantly, the antibody fusion reagent appears capable of tolerating lyophilization for ease of distribution. Furthermore, there is significant cost advantage to the HAT approach, as the large-scale production of the antibody fusion reagent (1-gram scale) is enough for 10 million tests (at ~0.27 UK pence/ < \$0.01 USD). The authors provide limited but encouraging direct point of care tests on capillary blood samples, which provides a roadmap of even more potential for this HAT strategy. Finally, the authors will ship lyophilized aliquots (good for ~10,000 tests, free of charge).

The limitations of their HAT approach, particularly the sensitivity (~90%) and specificity (99%), which do not fall completely above recommended government standards (98% - 99.5 % for both sensitivity and specificity), are clearly acknowledged, while providing a compelling argument for the use of HAT despite falling under these guidposts.

This work will certainly be of interest to the community and broader field alike. It is very likely to influence and shape low cost alternatives to develop serological COVID-19 tests and beyond, especially in low and middle-income countries.

Minor comments:

- The authors mention a group that has described preliminary results using an approach similar to that described here, except a scFV antibody against a different red blood cell H antigen was used. Might the authors provide some discussion on potential advantages of VHH domain vs. scFV (e.g., potential costs, storage, etc.)

Response:

There are many possible targets on the red cell surface that could be used to anchor the RBD. Some may be better than others. The advantage of the single domain VHH IH4 compared to a scFV is its simplicity and availability. It may be that Glycophorin A may be a better choice than H – the epitope is protein rather than carbohydrate; H deficiency in India has a frequency ~1:10,000 whereas GPA deficiency is said to be much rarer 1:1,000,000. However, any such comments are entirely speculative without appropriate comparative work. We have provided our IH4-RBD to Kruse et al (who described the scFV reagent) and their commercial partners for them to make some comparisons.

- Figure 2B: It is not clear if samples are duplicates.

Yes, these are duplicates. We have now clarified this in the figure legend (line 203).

- Page 7, line 177: The authors' state:" (Figure 2A Row 8)," why not refer to this the bottom, "Nil" row.

This has now been changed in the manuscript (line 171).

Reviewer #4 (Remarks to the Author):

A haemagglutination test for rapid detection of antibodies 1 to SARS-CoV-2

Alain Townsend^{1,2*}, Pramila Rijal^{1,2}, Julie Xiao¹, Tiong Kit Tan¹, Kuan-Ying A Huang³, Lisa Schimanski^{1,2}, Jiangdong Huo⁴, Nimesh Gupta⁵, Rolle Rahikainen⁶, Philippa C Matthews^{7,8}, Derrick Crook^{7,9,10}, Sarah Hoosdally^{8,10}, Teresa Street⁹, Justine Rudkin¹¹, Nicole Stoesser^{7,9}, Fredrik Karpe¹², Matthew Neville¹², Rutger Ploeg¹³, Marta Oliveira¹³, David J Roberts^{14,15}, Abigail A Lamikanra¹⁴, Hoi Pat Tsang¹⁴, Abbie Bown¹⁶, Richard Vipond¹⁶, Alexander JMentzer¹⁷, Julian C Knight¹⁷, Andrew Kwok¹⁷, Gavin Sreaton^{17,18}, JuthathipMongkolsapaya^{2,18,19}, Wanwisa Dejnirattisai¹⁷, Piyada Supasa¹⁷, Paul Klenerman⁸, ChristinaDold^{20,21}, Kenneth Baillie²², Shona C Moore²³, Peter JM Openshaw²⁴, Malcolm G Semple²³, Lance CW Turtle²⁵, Mark Ainsworth²⁶, Alice Allcock¹⁷, Sally Beer²⁶, Sagida Bibi²⁰, ElizabethClutterbuck²⁰, Alexis Espinosa²⁶, Maria Mendoza²⁶, Dominique Georgiou²⁶, Teresa Lockett²⁶, Jose Martinez²⁶, Elena Perez²⁶, Veronica Sanchez²⁶, Giuseppe Scozzafava²⁶, AlbertoSobrinodiaz²⁶, Hannah Thraves²⁶, Etienne Joly^{27*}

Townsend and colleagues present a haemagglutination assay for use in the detection of SARS-CoV-2 antibodies in serum/plasma. The assay consists of a nanobody-antigen fusion protein that effectively serves to cross-link or agglutinate red blood cells (RBCs) to one another in the presences of antibodies targeting the antigen of interest. In their assay system, the nanobody, VHH-1H4, targets Glycophorin A, a protein that is highly expressed on the

surface of RBCs. The nanobody is C-terminally fused to the receptor binding domain of the SARS-CoV-2 spike protein. Since antibodies are bivalent (i.e., IgG, IgA) or pentameric (IgM), one molecule is able to engage multiple RBC and high-order complexes are formed.

In this hemagglutination assay, they show that when RBCs from a naïve donor is mixed with their fusion protein and serum spiked with monoclonal antibodies or serum from COVID-19 subjects, the RBCs agglutinate in an anti-RBD antibody concentration dependent manner. The data in the manuscript shows that the test functions with monoclonal antibodies, convalescent serum and COVID-19 hospitalized patient serum obtained within the first 5 days of hospitalization. They also performed testing on serum obtained from patients from the pre-COVID-19 era to assay for the test's specificity. The authors report a sensitivity of ~90% and a specificity of ~99% in convalescent patients. The assay is relatively cheap ~£26 per plate. The benefit of this assay is that the assay requires little in the way of equipment, cold chain and molecular biology skills. However, I have significant concerns about the significance of this assay given its modest performance characteristics, other faster tests with similar performance (e.g., Binax Now) and the assay being incomplete (i.e., missing controls in the working setup and the lack of formal validation testing). This latter aspect is especially important as the authors mention multiple times how they are willing to provide materials to others, suggesting that the assay is ready for large scale use and it is not.

Response:

The referee has misread the cost of the HA test – the cost is 0.27 pence per test well ~ = 27 pence per plate = £0.27 pounds per plate. i.e. 100-fold less than the referee quotes.

The Referee also appears to have missed the fact that the BinaxNow (Cost \$5 = 378 pence = 1,400-fold more expensive than our HA test), is a test for SARS-CoV-2 antigen for identifying acutely infected individuals in the first week of infection, and our HAT detects antibodies to the SARS-CoV-2 RBD that develop in the weeks following infection. The two tests are not comparable and are used for completely different purposes. So, this comparison is not really appropriate. In addition, the company that market Binax Now quote sensitivity as low as 83% in samples with low levels of virus (<https://asm.org/Articles/2020/November/SARS-CoV-2-Testing-Sensitivity-Is-Not-the-Whole-St>). This needs to be confirmed by independent assessment. In the UK, Liverpool University found sensitivity as low as 48.8% for the similar Innova test in real application (https://assets.publishing.service.gov.uk/government/uploads/system/uploads/attachment_data/file/943187/S0925_Innova_Lateral_Flow_SARS-CoV-2_Antigen_test_accuracy.pdf). This could lead to disastrous misclassification of infectious individuals.

Response:

The referee claims our data are “missing controls in the working setup and the lack of formal validation testing” – this is palpably wrong as shown in the figures – we went to great lengths to obtain appropriate samples to validate the test, and repeated these experiments.

Specifically concerns include:

1) There is no data on RBC donor-to-donor variations or RBC age impacts the assay sensitivity and specificity.

Response:

The 232 samples from figure 5 were tested a second time 34 days later with a different red cell donor. 99% of the measured titres were within one doubling dilution of each other. We have measured the control antibody CR3022 with multiple donors and not detected more than 1 doubling dilution difference in titre.

Action:

In response to the objection raised by this referee, we have selected three additional O-ve donors and performed quadruplicate titrations of a) monoclonal antibodies to each structure defined binding site on the RBD (FI-3A Class 1, Huang 2020; C121, Class 2, Robbiani 2020; S309 Class 3, Pinto et al 2020; and CR3022 Class 4, Huo 2020), and b) three positive human convalescent sera with different titres, as well as a negative serum.

Results:

Maximal variation of +/- 1 doubling dilution between all of the measurements can be seen on slides 14-16 in the pdf file provided. Those have already been discussed earlier, in our response to reviewer 1, comment 4.

Manuscript:

This additional data is now included in Supplementary Tables 3 and 4, and described in the text Lines 278-284

2) Other than testing a set of pre-covid-19 serum samples, there is no formal evaluation on the impact of RBD targeting antibodies against endemic coronaviruses on the assay. Since the negative serum were not tittered against endemic coronaviruses, the specificity number may not be accurate.

Response:

This issue has already been discussed above, as an answer to the comment 3 from referee 1

- a. Similarly, ideally more than one antigen would be used targeting epitopes that are specific to SARS-CoV-2. This would be expected to increase sensitivity and perhaps specificity.

Response:

This is a request that would be difficult to fulfil, and for the reasons outlined above would in all likelihood be pointless. The other possible “targeting epitopes” such as the S2 domain of Spike and the N protein are the most cross-reactive so would very likely reduce the assay specificity.

3) While I applaud that the assay controls for the presence of anti-RBC targeting antibodies, there is a striking lack of a negative and positive control serum. Well characterized reagents for this purpose should be included in the assay on every plate. This can be a home-made or commercially provide reagent. If end-users are going to be expected to provide the material for their assay, the ideal performance characteristics of the controls should be well described, to allow for end-users to perform the assay appropriately.

Response:

The reviewer is mistaken, – we described multiple negative control monoclonal antibodies to other areas of the spike (anti NTD and anti S2), and there are 199 negative sera in figure 4

and 79 negative controls in Figure 5. We described multiple positive control monoclonal antibodies to the RBD. We also include a positive control Monoclonal called CR3022 that is extremely well characterised with a crystal structure, and for which we have described the performance in terms of titration in detail in the paper, and which we provide free in the assay reagents we are distributing as a positive control. This antibody has been titrated with red cells from several donors and gives highly consistent results. We do not feel it is necessary to include the CR3022 “on every plate” but should be included with a titration with any batch of plates, to ensure that the IH4-RBD reagent is functioning correctly, which we stipulate in the methods.

4) Early sensitivity is being defined as the first five days of hospitalization, but the data should be discussed in time post-symptom onset as the time to presentation is variable and has the potential to bias the sensitivity higher or lower.

Response:

We feel that have made this issue very clear in the paper on line 517. It is not always possible to define the date of onset of symptoms, but the date of admission is an objective reality. The key point is that the test frequently becomes positive during the first five days of an average admission in a UK hospital, and is at least as sensitive as the Siemens Atellica test for detection of antibodies to the RBD in this context.

5) Most critically, like other serologic assay’s its highest sensitivity is in convalescent patients and is not surprisingly least sensitive during early illness. This limit’s its use to population studies and for post-facto diagnosis of people who were not tested early on and is therefore, unlikely to have a significant impact on lowering disease transmission dynamics.

Response:

This statement is true of all serological testing, and does not warrant a detailed response.

1 Effect of Plasma v Serum on
Titration End-Points for
Reviewer 1, Comment 2

2. Summary

1. Sensitivity to agglutination by FI-3A (Class 1), C121 (Class 2), FD-11A (Class 3), and CR3022 (Class 4) Mabs is the same in Serum, K2EDTA Plasma or Heparin Plasma maintained at 1:20 dilution throughout.

Screening as in Fig 4 at 1:40 dilution can be done with serum or plasma samples.

2. End point Titres as in Fig 5 are the same from Serum and Plasma Samples. Compared to Antibodies starting in PBS The antibodies in Serum/Plasma in 2 of 4 cases titrate one DD further, but do not differ from each other.

3. Question: Does Serum v Plasma alter detection at the titration limit, as in the setting of screening at 1:40 in Fig 4? Total Vol 100ul.

- Take Blood into K2EDTA, Heparin, Clotted in Eppendorf. Allow clot to form 1hr RT, 1hr at 4° C
- Spin 5' Microfuge, decant plasma/serum
- Dilute to 1:20 in PBS
- Prepare Red cells taken in K2EDTA tube 1:20 in PBS
- Prepare FI-3A (Class 1), C121 (Class 2), FD-11A (Class 3) and CR3022 (Class 4) 20ug/ml in the 1:20 Serum/Plasmas or in PBS.
- Plate 25ul col 1, **prepare DD with the 1:20 Serum/Plasmas** Col 2-11, No Ab Col 12
- Add 25ul 1:20 Red Cells, Mix. Now serum is at 1:20 throughout with 1:40 red cells in vol 50ul as for set up to measure Sensitivity and Specificity, and in POC set up
- Add 50ul IH4-RBD (2ug/ml in PBS) = 100ng/well.
- Incubate 1 hr RT. Tilt 30s and Photograph

4

CR3022 Class 4 Monoclonal Ab ng/well

250 125 62.5 31.2 15.6 7.8 3.9 1.95 0.97 0.49 PBS

Serum
K2 EDTA Plasma
Heparin Plasma
PBS

End Points for Serum and Plasmas at 1:40 are the same

End Point in PBS = 0.97-1.95 ng/well

5

Repeat

CR3022 Class 4 Monoclonal Ab ng/well

250 125 62.5 31.2 15.6 7.8 3.9 1.95 0.97 0.49 PBS

End Points for Serum and Plasmas at 1:40 are the same

End Point in PBS = 0.97-1.95 ng/well

6

FI-3A Class 1 Monoclonal Ab ng/well

250 125 62.5 31.2 15.6 7.8 3.9 1.95 0.97 0.49 PBS

End Points for Serum and Plasmas at 1:40 are the same

End Point in PBS = 7.8 ng/well

7

C121 Class 2 Monoclonal Ab ng/well

250 125 62.5 31.2 15.6 7.8 3.9 1.95 0.97 0.49 PBS

End Points for Serum and Plasmas at 1:40 are the same

End Point in PBS = 7.8 ng/well

8

FD-11A Class 3 Monoclonal Ab ng/well

250 125 62.5 31.2 15.6 7.8 3.9 1.95 0.97 0.49 PBS

End Points for Serum and Plasmas at 1:40 are the same

End Point in PBS = 31.2 ng/well

9. Question: Does the origin of the sample alter the end-point titres, as in Figure 5? Mab plated in 50ul in 1:20 serum/plasma, *DD with 50ul PBS*, add 50ul 1:40 Red Cells, add 50 IH4-RBD (2ug/ml). Total Vol 150ul.

- Prepare Red cells taken in K2EDTA tube 1:40 in PBS
- Prepare MAbs at 20ug/ml in the 1:20 Serum/Heparin Plasma,/K2EDTA Plasma
- Plate 50ul col 1, prepare DD with PBS Col 2-11, No Ab Col 12
- Add 50ul 1:40 Red Cells, Mix.
- Add 50ul IH4-RBD (2ug/ml in PBS) = 100ng/well.
- Incubate 1 hr RT. Tilt 30s and Photograph

10

CR3022 Class 4 Monoclonal Ab ng/well

500 250 125 62.5 31.2 15.6 7.8 3.9 1.95 0.97 0.49 PBS

End Points for Serum and Plasmas at 1:40 are the same
End point in Sera/Plasma = 0.97 ng/well

End Point in PBS = 1.95 ng/well

11

FI-3A Class 1 Monoclonal Ab ng/well

500 250 125 62.5 31.2 15.6 7.8 3.9 1.95 0.97 0.49 PBS

End Points for Serum and Plasmas at 1:40 are the same

End Point in PBS = 7.8 ng/well

12

C121 Class 2 Monoclonal Ab ng/well

500 250 125 62.5 31.2 15.6 7.8 3.9 1.95 0.97 0.49 PBS

End Points for Serum and Plasmas at 1:40 are the same
One DD more sensitive
End Point = 3.9 ng/well

End Point in PBS
= 7.8 ng/well

13

FD-11A Class 3 Monoclonal Ab ng/well

500 250 125 62.5 31.2 15.6 7.8 3.9 1.95 0.97 0.49 PBS

End Points for Serum and Plasmas at 1:40 are the same

End Point in PBS = 31.2 ng/well

14. Variation between O-ve red cells from Different Donors and repeatability (replicates within the assay). Referee 1 , comment 4; Reviewer 4 Comment 1.

Method:

Three O -ve donors – 4ml Whole Blood taken in K3EDTA tube (PH: blood taken on 23 Nov 2020; JH and HM: blood taken on 26 Nov 2020)

RESULT: all end point titres are within +/- 1 DD of the central value regardless of observer or source of O-ve red cells

15

Monoclonal Antibodies FI-3A Class 1 (Huang 2020), C121 Class 2 (Robiani 2020), FD-11A Class 3 (Huang 2020) and CR3022 class 4 (Huo 2020) compared on 3 sources of O-ve Red cells (A-C) in quadruplicate (1-4), read by two independent Scorers.

	FI-3A Class 1			C121 Class 2			FD-11A Class 3			CR3022 Class 4			040 -ve Control anti Ebola		
O-ve Donr	A	B	C	A	B	C	A	B	C	A	B	C	A	B	C
1	8	9	8	7	8	7	4	5	4	8	9	8	0	0	0
2	8	10	8	7	8	7	4	5	4	8	9	8	0	0	0
3	8	9	8	7	8	7	4	4	4	8	9	8	0	0	0
4	8	9	8	7	8	7	4	4	4	8	9	8	0	0	0

	FI-3A Class 1			C121 Class 2			FD-11A Class 3			CR3022 Class 4			040 -ve Control anti Ebola		
O-ve Donr	A	B	C	A	B	C	A	B	C	A	B	C	A	B	C
1	8	9	8	7	8	7	4	5	4	8	9	8	0	0	0
2	8	9	8	7	8	7	4	5	4	8	9	8	0	0	0
3	8	9	8	7	8	7	4	5	4	8	9	8	0	0	0
4	8	9	8	7	8	7	4	4	4	8	9	8	0	0	0

16

Four plasma samples selected for high titre (9DD = 1:10,240), Medium Titre (6DD = 1: 1,280), Low Titre (3DD – 1:160) and a negative serum titrated on 3 sources of O-ve Red cells (A-C) in quadruplicate (1-4), read by two independent scorers.

	Serum 9A			Serum 11B			Serum 11G			Serum 10G -ve Control		
O-ve Donor:	A	B	C	A	B	C	A	B	C	A	B	C
1	9	9	9	5	7	6	3	4	3	0	0	0
2	9	9	9	5	7	6	3	4	3	0	0	0
3	9	9	9	6	7	6	3	4	3	0	0	0
4	9	9	9	5	7	6	3	4	3	0	0	0
	Serum 9A			Serum 11B			Serum 11G			Serum 10G -ve Control		
O-ve Donor:	A	B	C	A	B	C	A	B	C	A	B	C
1	9	9	9	5	7	6	3	4	3	0	0	0
2	9	9	9	5	7	6	3	4	3	0	0	0
3	9	9	9	5	6	6	3	4	3	0	0	0
4	9	9	9	5	6	6	3	4	3	0	0	0

Reviewer #2 (Remarks to the Author):

Hello, I found the authors rebuttal to the reviewer comments to be very extensive and thorough, addressing the points raised. This included both comments for my questions as well as the other reviewer comments. While there are more work to be done with this technology to bring to clinical utilization, the work provides very thorough evaluation of the proof of principle with many clinical samples on its potential use. I have no further critiques and comments at this time.

Reviewer #3 (Remarks to the Author):

In their revision, the authors have sufficiently addressed my minor comments/concerns.

Reviewer #4 (Remarks to the Author):

A haemagglutination test for rapid detection of antibodies 1 to SARS-CoV-2
Alain Townsend, et al

[removed for brevity] The authors report a sensitivity of ~90% and a specificity of ~99% in convalescent patients. The assay is relatively cheap ~£26 per plate. The benefit of this assay is that the assay requires little in the way of equipment, cold chain and molecular biology skills. However, I have significant concerns about the significance of this assay given its modest performance characteristics, other faster tests with similar performance (e.g., Binax Now) and the assay being incomplete (i.e., missing controls in the working setup and the lack of formal validation testing). This latter aspect is especially important as the authors mention multiple times how they are willing to provide materials to others, suggesting that the assay is ready for large scale use and it is not.

[Author] Response:

The referee has misread the cost of the HA test – the cost is 0.27 pence per test well ~ = 27 pence per plate = £0.27 pounds per plate. i.e. 100-fold less than the referee quotes. The Referee also appears to have missed the fact that the BinaxNow (Cost \$5 = 378 pence = 1,400-fold more expensive than our HA test), is a test for SARS-CoV-2 antigen for identifying acutely infected individuals in the first week of infection, and our HAT detects antibodies to the SARS-CoV-2 RBD that develop in the weeks following infection. The two tests are not comparable and are used for completely different purposes. So, this comparison is not really appropriate. In addition, the company that market Binax Now quote sensitivity as low as 83% in samples with low levels of virus (<https://asm.org/Articles/2020/November/SARS-CoV-2-Testing-Sensitivity-Is-Not-the-Whole-St>). This needs to be confirmed by independent assessment. In the UK, Liverpool University found sensitivity as low as 48.8% for the similar Innova test in real application (https://assets.publishing.service.gov.uk/government/uploads/system/uploads/attachment_data/file/943187/S0925_Innova_Lateral_Flow_SARS-CoV-2_Antigen_test_accuracy.pdf). This could lead to disastrous misclassification of infectious individuals.

Reviewer 4 comment on response:

- The reviewer did indeed misread the cost. Nonetheless, the reviewer indicated that it was relatively cheap and was not making the comparison to the cost of the test but rather the other parameters of the assays (i.e., sensitivity and specificity). That said, that standard for clinical testing would not allow for reuse of plates, therefore that should not be assumed to be a reasonable expectation in cost calculations. Further, running one sample would use one plate. Based on this, a range of cost should be included to reflect the cost of an entire plate and other materials that would be needed to run the assay. I agree with the authors, staff labor and blood drawing costs should not be included. I would estimate it is in the \$2 to \$5 range, but the authors

should come up with their own estimate. The authors should explicitly state that the costs do not include labor.

- I agree that Binax Now and other tests do have variable sensitivities compared to the manufacturer's stated claims once the tests leave the lab and hit real world scenarios. Binax now was but one example and was listed as one example but was not meant to be a specific comparison. There are others assays that can be used as examples ([https://www.thelancet.com/journals/lancet/article/PIIS0140-6736\(20\)32635-0/fulltext](https://www.thelancet.com/journals/lancet/article/PIIS0140-6736(20)32635-0/fulltext)). The concern is that as a point of care diagnostic test, the HAT assay does not show a significant advantage over commercially available tests that are clinically validated and are subject to rigorous quality control standards.
- More importantly, the authors make my point that using any test with poor sensitivity can have large impact on public health outcomes (<https://asm.org/Articles/2020/November/SARS-CoV-2-Testing-Sensitivity-Is-Not-the-Whole-St>) both on the positive and negative predictive values depending on the prevalence of COVID-19 in the population being tested (<https://www.ncbi.nlm.nih.gov/pmc/articles/PMC5701930/>, <https://www.ons.gov.uk/peoplepopulationandcommunity/healthandsocialcare/conditionsanddiseases/bulletins/coronaviruscovid19infectionsurveyspilot/18december2020>) . The sensitivity in the hospital setting for the HAT assay in one test sample was 86% and specificity was 100%. It is unrealistic to expect that the 100% specificity will hold up in the real world for this assay. If the prevalence is 1.25 % (i.e., the estimate England prevalence Day 5-18, <https://www.ons.gov.uk/peoplepopulationandcommunity/healthandsocialcare/conditionsanddiseases/bulletins/coronaviruscovid19infectionsurveyspilot/24december2020>) and the specificity is lowered to 99.5%, the positive predictive value of the test is only 68% and gets worse as the specificity gets lower (i.e., 52% with specificity of 99%; 42% with specificity of 98.5%). To mitigate the poor positive predictive value, one would prefer a test with higher sensitivity as a point of care test in the hospital setting. Because this assay (and other assays) does not provide a benefit and should not be used for this purpose.
- This test is therefore only of potential use as an epidemiologic tool to replace standard immunologic assays like ELISA tests. Since the clinical utility of the assay is low the overall significance of this assay is also low.

[Author] Response:

The referee claims our data are "missing controls in the working setup and the lack of formal validation testing" – this is palpably wrong as shown in the figures – we went to great lengths to obtain appropriate samples to validate the test, and repeated these experiments. 

Reviewer 4 comment on response:

Controls: A dilution series are more informative and reliable for the identification of positivity when performing assays as they provide a titer value. While single point assays do have a suggested control, the method described for the dilution series assay does not include controls for that assay and controls do not appear in Figure 5, which was the assay shown for the real-world samples and presumably is the recommended plate layout setup. The appropriateness of choice of controls used in other assay methods and referred to by the authors is discussed later.

Formal Validation testing: This assay is not a simple description of a method, but strongly advocates at several points for its use in a clinical setting in resource limited settings. Assays being advocated for resource limited settings should be held to the same standards as what would be expected in the 1st world. The standard this clinical assay should therefore be formal validation with quality standards defined and presented prior to being advocated for broad use. A good example of an assay that was developed for use in both 1st world and resource limited setting that underwent validation and has standardized controls is the FANG Ebolavirus glycoprotein ELISA assay for human samples (<https://journals.plos.org/plosone/article?id=10.1371/journal.pone.0215457>) .

The HAT assay lacks “formal validation testing”, including the definition of assay acceptance criteria, determination of the impact of interfering agents in serum (e.g., hemoglobin, albumin, triglycerides), determination of operator variance (i.e., running of the assays, not just interpretation of the results), the lower limit of quantification of the assay using serum and a formal assessment of the stability of reagents being supplied, stability/handling/storage of RBCs and repeating the assays on the same samples on different days and by different operators. Since the authors report that the assay is rapid and high throughput, they should be able to perform many/all of these assays or they should remove all references to ease of shipping, costs of the protein and the willingness to ship the material to any lab that wants it.

Specifically concerns include:

1) There is no data on RBC donor-to-donor variations or RBC age impacts the assay sensitivity and specificity.

[Author] Response:

The 232 samples from figure 5 were tested a second time 34 days later with a different red cell donor. 99% of the measured titres were within one doubling dilution of each other. We have measured the control antibody CR3022 with multiple donors and not detected more than 1 doubling dilution difference in titre.

Action:

In response to the objection raised by this referee, we have selected three additional O-ve donors and performed quadruplicate titrations of a) monoclonal antibodies to each structure defined binding site on the RBD (FI-3A Class 1, Huang 2020; C121, Class 2, Robbiani 2020; S309 Class 3, Pinto et al 2020; and CR3022 Class 4, Huo 2020), and b) three positive human convalescent sera with different titres, as well as a negative serum.

Results:

Maximal variation of +/- 1 doubling dilution between all of the measurements can be seen on slides 14-16 in the pdf file provided. Those have already been discussed earlier, in our response to reviewer 1, comment 4.

Manuscript:

This additional data is now included in Supplementary Tables 3 and 4, and described in the text Lines 278-284

Reviewer 4 comment on response:

RBC age. The response does not include any assessment on RBC age as requested. That is, there is no description of whether fresh RBC's are required. The manuscript comments on how many wells can be assayed using a single isolation of RBCs from whole blood but lacks this critical piece of information.

Donor to donor variation. The data presented partially addresses the concern. What is lacking is an indication of whether the quadruplicate experiments were presented independent replicates, done on the same day or the number of operators who performed the assay (not just the interpretation).

2) Other than testing a set of pre-covid-19 serum samples, there is no formal evaluation on the impact of RBD targeting antibodies against endemic coronaviruses on the assay. Since the negative serum were not tittered against endemic coronaviruses, the specificity number may not be accurate.

[Author] Response:

This issue has already been discussed above, as an answer to the comment 3 from referee 1

[Author] Response [comment 3, referee 1]:

We tested 54 samples from patients with sepsis, all of which tested negative, and a total of 224 samples from other control donors giving a total of 278 negative samples with a single weak false positive result in the assays to assess sensitivity and specificity. Many of these donors are expected to have had high titre antibodies to influenza and other common infections, including common cold coronaviruses. Since those were clearly not detected in the test, we do feel that we have shown a reasonable data set to establish that, if there is some cross reactivity to other viruses, it is a very infrequent problem for HAT, and that the test is at least 99% specific. To do as the referee asks would require a properly designed prospective study to collect blood samples from a large cohort of patients with respiratory illness for whom gold standard PCR diagnosis of each infection was available. This is simply not possible for this paper.

As a further elaboration on this issue, we know from published data that ~90% of adults are seropositive for OC43 (the most closely related of the common cold Betacoronaviruses), (Severance et al; <https://www.ncbi.nlm.nih.gov/pmc/articles/PMC2593164/>; and Anderson et al BioRxiv 2020, <https://www.medrxiv.org/content/10.1101/2020.11.06.20227215v1>), and yet we are not detecting a high false positive rate in the pre-covid samples. This is because the great majority of this cross-reactivity is directed at the N protein (Severance et al) and the S2 domain of the Spike protein (Ng et al, <https://science.sciencemag.org/content/early/2020/11/05/science.abe1107>);). In the latter paper all of the cross-reactivity to the spike protein was absorbed out with the isolated S2 domain, but NOT by the S1 domain containing the RBD. This is very likely because there is very much less conservation of the RBD sequence between these viruses (~22% v 43% for the S2 domain). In our own work characterising monoclonal antibodies to SARS-CoV-2 (Huang et al, <https://www.biorxiv.org/content/10.1101/2020.08.28.267526v1>;) we found 5 out of 9 MAbs to S2 contained sequence features of memory cells (high level of somatic mutation) and cross-reacted on OC43 S2, and a similar proportion for N. For the RBD we found just 1 of 31 MAbs that cross-reacted with RBD from OC43. This single antibody was close to germline in sequence (i.e. Not derived from a memory B cell), and in any case failed to cross-link red cells in the HA test.

Finally, in our two assays to measure the operating characteristics of the HA test, the single false positive for each run was derived from a single well that showed a partial teardrop in one assay (scored negative) and loss of teardrop in the second (scored positive). In each case we investigated these by repeat measurements, and a titration with a reagent composed of IH4 without the RBD. In both cases the repeat assays showed only a partial teardrop in the first well of the titration, showing that whatever antibody was causing the reaction was low titre. In one case this weak reaction was clearly dependent on the presence of the RBD, i.e. could have been due to rare and very low titre cross-reactive antibodies between the RBD of a common cold virus and SARS-CoV-2. This is consistent with detection in 0.6-2% of precovid donors by Anderson et al. (see above) of binding activity to the RBD of SARS-CoV-2. In the second case the reaction was with the IH4 component. Lastly, we have heard from our colleague Prof Gathsaurie Malavige in Sri Lanka, who has been using the HA test successfully for some time, and found a single false positive in 110 pre-covid samples, from a patient with Dengue.

So, with this evidence in hand, we feel that false positive results in the HA test due to very low-level cross-reactivity between the RBDs of common cold betacoronaviruses and the RBD of SARS-CoV-2 will be rare (less than 1%), and can be distinguished by their low titre.

Reviewer 4 comment on response:

The betacoronaviruses HKU1 and OC43 are most closely related to SARS COV-2. However, exposure to the four endemic coronaviruses (i.e., 229E, HKU1, NL63 and OC43) in adults is high and the papers the authors refer to indicate that subjects have 60-92% seropositivity to the nucleoprotein for these species. While the data referenced by the authors shows a low level of cross-reactivity to SARS CoV-2 Spike in pre-pandemic serum, without testing the serum for activity against SARS CoV2 or endemic coronavirus, one should not assume the pre-pandemic

serum is in and of itself a sufficient test. One particular concern would be an individual who has a high-titer to an endemic coronavirus spike protein following a recent infection and has developed high cross-reactivity to SARS CoV-2 Spike. This might cause a false positive. To test this, the authors should have tested pre-pandemic sera with high reactivity to the spike protein of each of the four human endemic coronaviruses. The susceptibility of a diagnostic assay to conditions with the potential to impact its performance are important to define and understand prior to being broadly adapted. The requested analysis was not performed and therefore the authors did not respond this reviewer's concern.

a. Similarly, ideally more than one antigen would be used targeting epitopes that are specific to SARS-CoV-2. This would be expected to increase sensitivity and perhaps specificity.

[Author] Response:

This is a request that would be difficult to fulfil, and for the reasons outlined above would in all likelihood be pointless. The other possible "targeting epitopes" such as the S2 domain of Spike and the N protein are the most cross-reactive so would very likely reduce the assay specificity.

Reviewer 4 comment on response:

The authors response discusses the rationale for choosing RBD over other domains in Spike and using other virus protein in the response to reviewer #1, comment 3. The manuscript would be improved by adding this information and rationale to the manuscript.

3) While I applaud that the assay controls for the presence of anti-RBC targeting antibodies, there is a striking lack of a negative and positive control serum. Well characterized reagents for this purpose should be included in the assay on every plate. This can be a home-made or commercially provide reagent. If end-users are going to be expected to provide the material for their assay, the ideal performance characteristics of the controls should be well described, to allow for end-users to perform the assay appropriately.

[Author] Response:

The reviewer is mistaken, – we described multiple negative control monoclonal antibodies to other areas of the spike (anti NTD and anti S2), and there are 199 negative sera in figure 4 and 79 negative controls in Figure 5. We described multiple positive control monoclonal antibodies to the RBD. We also include a positive control Monoclonal called CR3022 that is extremely well characterised with a crystal structure, and for which we have described the performance in terms of titration in detail in the paper, and which we provide free in the assay reagents we are distributing as a positive control. This antibody has been titrated with red cells from several donors and gives highly consistent results. We do not feel it is necessary to include the CR3022 "on every plate" but should be included with a titration with any batch of plates, to ensure that the IH4-RBD reagent is functioning correctly, which we stipulate in the methods.

Reviewer 4 comment on response:

While the monoclonal antibodies (mAbs) and serum used to develop the HAT are important, they are not used appropriately in the final assay as controls. First, mAbs are used in PBS and not in serum or plasma. This is not appropriate because the matrix can have an impact on the assay due to variations in serum/plasma proteins, fat content, hemolysis or other components present that are not in PBS. These physiochemical properties could impact the binding properties of the assay.

Next, the authors note the protein is stable following reconstitution for 1-2 weeks when refrigerated, there is no data presented on how that was determined and what the long-term stability is of the lyophilized material. Therefore, it is important to have a control that can accurately monitor for the loss of IH4-RBD integrity. In the single point and POCT assays it appears that the methods suggest using only one well to investigate the IH4-RBD integrity. It

should be done as a dilution series in order to assess for a loss in antigen potency. The expected potency for the CR3022 in the assay should be provided as part of the acceptance criteria for the assay. Without this, the sensitivity of the assay might be impacted by IH4-RBD protein that has been degraded due to improper storage, handling or reconstitution by the operator.

Furthermore, while mAbs are easier to supply as a reagent they are not the ideal control for an assay. Ideally, a high and low positive control serum and negative control serum would be provided as a reagent and used as a dilution series in the assay. This will provide a positive control for the reagents and be used to confirm that the assay is performing as expected (i.e., within acceptance criteria). This is missing from that assay. Alternatively, mAb-spiked serum could be provided.

Additionally, ideally each plate would have a control associated with it to account for potential plate to plate variability and to confirm that the operator performed that assay as expected on each plate.

Finally, the titration method (page 25#2) does not mention a batch CR3022 control and, as currently written, also does not test for the presence of cold agglutinins by mixing serum with PBS and RBCs in the absence of RBD.

4) Early sensitivity is being defined as the first five days of hospitalization, but the data should be discussed in time post-symptom onset as the time to presentation is variable and has the potential to bias the sensitivity higher or lower.

[Author] Response:

We feel that have made this issue very clear in the paper on line 517. It is not always possible to define the date of onset of symptoms, but the date of admission is an objective reality. The key point is that the test frequently becomes positive during the first five days of an average admission in a UK hospital, and is at least as sensitive as the Siemens Atellica test for detection of antibodies to the RBD in this context.

Reviewer 4 comment on response:

While the objective reality of an admission date is certainly easy to ascertain, the conclusions that are being drawn are potentially confounded by inaccurately using the date of admission. Presumably each of the subjects has a history and physical exam that was written at the time of admission to the hospital. Any good clinician would have asked their patients when they started to get ill and would have noted it in the associated NHS medical record. This should be easily obtained by a chart review and the data is likely to be available for the vast majority of subjects.

In the absence of reporting the data as requested, the authors should emphasize this a second time when discussion their conclusions (i.e., not just at the start of the section).

5) Most critically, like other serologic assay's its highest sensitivity is in convalescent patients and is not surprisingly least sensitive during early illness. This limit's its use to population studies and for post-facto diagnosis of people who were not tested early on and is therefore, unlikely to have a significant impact on lowering disease transmission dynamics.

[Author] Response:

This statement is true of all serological testing, and does not warrant a detailed response.

Reviewer 4 comment on response:

The authors agree that the utility of this assay is in population studies and post-facto diagnosis and not as a test in the acute illness stage. Given this, the HAT in its current form has a low significance to the field.

Referee 2 (notes on report by referee 4)

It is clear R4 has taken a very careful look with the eyes of gold-standard assay validation. I just don't know if it is the right place to apply it in this instance.

With my reading this was not a clinical assay. Even if the assay is just used as an epidemiologic tool, it is still useful (also agreed to by R4). I do not agree (as stated by R4) that if the clinical utility of assay is potentially lower, the significance of this assay is low. This assay can help assess post infection assessment and vaccine development providing data that may otherwise never be collected (due to the low cost). In fact, when I read the paper, I was particularly pleased with (1) the low cost/simplicity and (2) free distribution to interested parties. The latter of which I imagined would lead to additional collaborations, assessments, and advancements of the assay. This is a free assay and in my opinion the benefit to using it is worth it.

Much of R4's issues appear to revolve around use in clinical setting. Ultimately, this seems to be more of philosophical debate: must assays that may have use in areas of limited resource be held to same standards of "1st world" settings.

- Perhaps the authors might lessen their suggestions of use or state due to the unique nature of the pandemic, validation will continue (basically a disclaimer). However, I feel I fall on the side of wider access to such assays, especially due to the severe limitations of any "perfect" assay and its likely much more limited distribution. While I am not directly in the clinical/diagnostic assay field, it is my understanding that a lot of diagnostic efforts to help low resource regions rely on techniques that may be less than gold standard assays.
- Perhaps the authors might note R4 concerns, such as RBC donor-to-donor variations, and then address in follow up studies.
- Similar noting cross-reactivity mentioned by R4, and acknowledging to address in follow up studies.

Reasonable requests.

- R4's request for controls for dilution series experiments is reasonable if it will be the recommended setup. Similar, authors should indicate details of R4's comments on donor-to-donor variation.
- R4's request to discuss rationale for choosing RBD
- R4's following request are reasonable (suggested):

While the monoclonal antibodies (mAbs) and serum used to develop the HAT are important, they are not used appropriately in the final assay as controls. First, mAbs are used in PBS and not in serum or plasma. This is not appropriate because the matrix can have an impact on the assay due to variations in serum/plasma proteins, fat content, hemolysis or other components present that are not in PBS. These physiochemical properties could impact the binding properties of the assay.

Next, the authors note the protein is stable following reconstitution for 1-2 weeks when refrigerated, there is no data presented on how that was determined and what the long-term stability is of the lyophilized material. Therefore, it is important to have a control that can accurately monitor for the loss of IH4-RBD integrity. In the single point and POCT assays it appears that the methods suggest using only one well to investigate the IH4-RBD integrity. It should be done as a dilution series in order to assess for a loss in antigen potency. The expected potency for the CR3022 in the assay should be provided as part of the acceptance criteria for the assay. Without this, the sensitivity of the assay might be impacted by IH4-RBD protein that has been degraded due to improper storage, handling or reconstitution by the operator.

Furthermore, while mAbs are easier to supply as a reagent they are not the ideal control for an

assay. Ideally, a high and low positive control serum and negative control serum would be provided as a reagent and used as a dilution series in the assay. This will provide a positive control for the reagents and be used to confirm that the assay is performing as expected (i.e., within acceptance criteria). This is missing from that assay. Alternatively, mAb-spiked serum could be provided.

Additionally, ideally each plate would have a control associated with it to account for potential plate to plate variability and to confirm that the operator performed that assay as expected on each plate.

Finally, the titration method (page 25#2) does not mention a batch CR3022 control and, as currently written, also does not test for the presence of cold agglutinins by mixing serum with PBS and RBCs in the absence of RBD.

Referee 3 (comments on report by referee 4)

Comment 1:

I think a compromise could be reached to state that the reagent cost is 0.27 pence, and that clinical use could be \$2-5, including blood draw, staff labor, and disposable plates.

-the next comment here kind of misses the point; this HAT test doesn't need to be better than the commercial tests, just cheaper

-the comments about sensitivity/specificity of serology tests in general are outside the scope of this manuscript. All tests suffer from limitations of positive predictive value, so I don't really get the relevance of arguing/debating that in this manuscript, which focuses on a novel technology, with great low cost.

-The reviewer arguing about the clinical utility of serology tests is again missing the point of the paper, and outside the scope of the manuscript. The reality is that antibody testing has proven useful for judging who may be protected from infection, who may benefit from convalescent plasma or recombinant monoclonal antibody therapy, who may not pose an infectious risk during COVID-19 infection and help prioritize vaccination resources. It's not simply epidemiology. Lots of people are getting ELISA tests for antibodies at this moment, and those are costing a significant amount of money, so it's not just for epidemiology.

Again, in my opinion, saying that antibody tests are stupid so this paper is stupid is more of a personal opinion, and not relevant to the data findings in the work.

Comment 2:

I thought the controls were appropriate and do not have issue with them. They did rigorous sensitivity and specificity testing with clinical samples, and in the earlier work used PBS controls. Clearly the assay works as intended and the authors don't need to repeat the same controls in every figure.

I don't think this HAT assay is meant to be shippable tomorrow as a clinical test. The paper focuses on characterizing the technology and presents data suggesting clinical utility. Subsequent testing will enable this to occur. Remember, this is an academic lab publishing this, not a diagnostic testing company. The Nature communications paper isn't an FDA submission packet. I think it's outside the scope of the authors expertise.

Comment 3:

I think the RBC age question is kind of stupid. The actual clinical assay would be using dilute whole blood from donors anyways. The authors used serum and RBC's from various donors mostly because only COVID-19 serum is banked right now. Any blood bank reagent red cells used furthermore are already a sold product, so this isn't something that needs to be reinvented.

I'm failing to see the concern of the reviewer on donor to donor variation. The fusion protein targets a conserved epitope in the human population. Not sure why the reviewer is worried? Seems like asking questions to ask questions.

Comment 4:

The reviewer appears to not know the current literature. There is a lot of data investigating cross-reactivity of RBD against the other endemic viruses (229E, HKU1, NL63 and OC43). Stating some hypotheticals by high titer cross-reactivity has not been born out in any of the other serology papers I've seen, including the Nature Medicine one. So again, this suggestion seems to be pure speculation with no literature to back it up, and I'm not even sure how the authors would obtain it. Moreover, this risk is for all serology tests, so doesn't need to be litigated by this one group.

<https://www.nature.com/articles/s41591-020-0913-5>

Comment 5:

I think a compromise would be add the rationale for choosing RBD. Would just require 1-2 sentences and a couple of references to the paper.

Comment 6:

I think this debate is sort of about the potential commercial rollout of the test, and what the best positive control for such a test to be. Arguing about this is kind of futile, since it will be cost considerations for any company that picks this up for formal regulatory approval. It might be lost on reviewer #4 that academic labs would benefit from a cheap test to do titration assays for COVID-19 samples in a research setting. In any case, a question of cost/money and exact controls for the commercial test shouldn't hold up publication.

I think the authors could add one more sentence to their protocol about having a control without RBD fusion protein, and just PBS and patient RBC/serum.

Comment 7:

I would disagree that it's easy to always know from patient data when the exact symptoms start. I am a clinician and have discussed with many colleagues, including friends and family, that it was vague when they might have started feeling bad. Even if one did do chart review, it would take a long time to acquire all of this information, and it may still be inaccurate. I sympathize more with the authors here, that using known admission dates is an objective criteria. Moreover, the time of seroconversion has already been studied in other manuscripts, and is outside the focus of the current technical manuscript. I think the authors noting this limitation once was enough.

Comment 7:

I'm not sure why this reviewer has a very narrow mind when it comes to the utility of serology tests. The lack of widely available serology tests has been a problem during the pandemic, preventing their application for different stages and/or aspects of treatment. Stating an entire testing category has low significance is an opinion to me. If such low significance, why did Nature Medicine publish an entire paper on an ELISA assay.

REVIEWER COMMENTS

Reviewer #2 (Remarks to the Author):

Hello, I found the authors rebuttal to the reviewer comments to be very extensive and thorough, addressing the points raised. This included both comments for my questions as well as the other reviewer comments. While there are more work to be done with this technology to bring to clinical utilization, the work provides very thorough evaluation of the proof of principle with many clinical samples on its potential use. I have no further critiques and comments at this time.

Reviewer #3 (Remarks to the Author):

In their revision, the authors have sufficiently addressed my minor comments/concerns.

We thank the referees 2 &3 wholeheartedly for their appraisal of our work, and even more so for the time and effort they dedicated to the additional evaluation of the flood of critical comments made by referee 4.

Reviewer #4 (Remarks to the Author):

A haemagglutination test for rapid detection of antibodies 1 to SARS-CoV-2

Alain Townsend, et al

[removed for brevity] The authors report a sensitivity of ~90% and a specificity of ~99% in convalescent patients. The assay is relatively cheap ~£26 per plate. The benefit of this assay is that the assay requires little in the way of equipment, cold chain and molecular biology skills. However, I have significant concerns about the significance of this assay given its modest performance characteristics, other faster tests with similar performance (e.g., Binax Now) and the assay being incomplete (i.e., missing controls in the working setup and the lack of formal validation testing). This latter aspect is especially important as the authors mention multiple times how they are willing to provide materials to others, suggesting that the assay is ready for large scale use and it is not.

[Author] Response:

The referee has misread the cost of the HA test – the cost is 0.27 pence per test well \approx 27 pence per plate = £0.27 pounds per plate. i.e. 100-fold less than the referee quotes. The Referee also appears to have missed the fact that the BinaxNow (Cost \$5 = 378 pence = 1,400-fold more expensive than our HA test), is a test for SARS-CoV-2 antigen for identifying acutely infected individuals in the first week of infection, and our HAT detects antibodies to the SARS-CoV-2 RBD that develop in the weeks following infection. The two tests are not comparable and are used for completely different purposes. So, this comparison is not really appropriate. In addition, the company that market Binax Now quote sensitivity as low as 83% in samples with low levels of virus (<https://asm.org/Articles/2020/November/SARS-CoV-2-Testing-Sensitivity-Is-Not-the-Whole-St>). This needs to be confirmed by independent assessment. In the UK, Liverpool University found sensitivity as low as 48.8% for the similar Innova test in real application (https://assets.publishing.service.gov.uk/government/uploads/system/uploads/attachment_data/file/943187/S0925_Innova_Lateral_Flow_SARS-CoV-2_Antigen_test_accuracy.pdf). This could lead to disastrous misclassification of infectious individuals.

Reviewer 4 comment on response:

• The reviewer did indeed misread the cost. Nonetheless, the reviewer indicated that it was relatively cheap and was not making the comparison to the cost of the test but rather the other parameters of the assays (i.e., sensitivity and specificity). That said, that standard for clinical testing would not allow for reuse of plates, therefore that should not be assumed to be a reasonable expectation in cost calculations. Further, running one sample would use one plate. Based on this, a range of cost should be included to reflect the cost of an entire plate and other materials that would be needed to run the assay. I agree with the authors, staff labor and blood drawing costs should not be included. I would estimate it is in the \$2 to \$5 range, but the authors should come up with their own estimate. The authors should explicitly state that the costs do not include labor.

A sentence pertaining to the overall cost of consumables being less than 1 £/test has been added (line 495)

• I agree that Binax Now and other tests do have variable sensitivities compared to the manufacturer's stated claims once the tests leave the lab and hit real world scenarios. Binax now was but one example and was listed as one example but was not meant to be a specific comparison. There are others assays that can be used as examples ([https://www.thelancet.com/journals/lancet/article/PIIS0140-6736\(20\)32635-0/fulltext](https://www.thelancet.com/journals/lancet/article/PIIS0140-6736(20)32635-0/fulltext)). The concern is that as a point of care diagnostic test, the HAT assay does not show a significant advantage over commercially available tests that are clinically validated and are subject to rigorous quality control standards.

We have tried to clarify in the paper that HAT is not initially intended for clinical use. The HAT has the important advantage of being free for the reagent as we are offering to supply it free of charge and as needed to any interested research group. Finally, concerning validation – the HAT can be controlled and standardised very accurately by comparison with the now available standard sera from NIBSC and WHO. We have now included comments on this in the manuscript (lines 286-291-, 516-519 & 711-713).

• More importantly, the authors make my point that using any test with poor sensitivity can have large impact on public health outcomes

(<https://asm.org/Articles/2020/November/SARS-CoV-2-Testing-Sensitivity-Is-Not-the-Whole-St>) both on the positive and negative predictive values depending on the prevalence of COVID-19 in the population being tested

(<https://www.ncbi.nlm.nih.gov/pmc/articles/PMC5701930/>, <https://www.ons.gov.uk/peoplepopulationandcommunity/healthandsocialcare/conditionsanddiseases/bulletins/coronaviruscovid19infectionsurveyspilot/18december2020>) . The sensitivity in the hospital setting for the HAT assay in one test sample was 86% and specificity was 100%. It is unrealistic to expect that the 100% specificity will hold up in the real world for this assay. If the prevalence is 1.25 % (i.e., the estimate England prevalence Day 5-18,

<https://www.ons.gov.uk/peoplepopulationandcommunity/healthandsocialcare/conditionsanddiseases/bulletins/coronaviruscovid19infectionsurveyspilot/24december2020>) and the specificity is lowered to 99.5%, the positive predictive value of the test is only 68% and gets worse as the specificity gets lower (i.e., 52% with specificity of 99%; 42% with specificity of 98.5%). To mitigate the poor positive predictive value, one would prefer a test with higher sensitivity as a point of care test in the hospital setting. Because this assay (and other assays) does not provide a benefit and should not be used for this purpose.

We have tried to clarify in the paper that HAT is not initially intended for clinical use. No serological assay can be expected to have high sensitivity early in the clinical course.

• This test is therefore only of potential use as an epidemiologic tool to replace standard immunologic assays like ELISA tests. Since the clinical utility of the assay is low the overall significance of this assay is also low.

This comment is unwarranted. Our proposal is to make HAT available as a test that can replace expensive ELISAs for those countries that cannot afford ELISA tests for sero-prevalence research studies.

[Author] Response:

The referee claims our data are “missing controls in the working setup and the lack of formal validation testing” – this is palpably wrong as shown in the figures – we went to great lengths to obtain appropriate samples to validate the test, and repeated these experiments.

Reviewer 4 comment on response:

Controls: A dilution series are more informative and reliable for the identification of positivity when performing assays as they provide a titer value. While single point assays do have a suggested control, the method described for the dilution series assay does not include controls for that assay and controls do not appear in Figure 5, which was the assay shown for the real-world samples and presumably is the recommended plate layout setup. The appropriateness of choice of controls used in other assay methods and referred to by the authors is discussed later.

We are at a loss to understand this comment since we do give titrations in these figures.

Formal Validation testing: This assay is not a simple description of a method, but strongly advocates at several points for its use in a clinical setting in resource limited settings. Assays being advocated for resource limited settings should be held to the same standards as what would be expected in the 1st world. The standard this clinical assay should therefore be formal validation with quality standards defined and presented prior to being advocated for broad use. A good example of an assay that we developed for use in both 1st world and resource limited setting that underwent validation and has standardized controls is the FANG Ebolavirus glycoprotein ELISA assay for human samples (<https://journals.plos.org/plosone/article?id=10.1371/journal.pone.0215457>).

We have tried to clarify in the paper that HAT is not initially intended for clinical use

The HAT assay lacks “formal validation testing”, including the definition of assay acceptance criteria, determination of the impact of interfering agents in serum (e.g., hemoglobin, albumin, triglycerides), determination of operator variance (i.e., running of the assays, not just interpretation of the results), the lower limit of quantification of the assay using serum and a formal assessment of the stability of reagents being supplied, stability/handling/storage of RBCs and repeating the assays on the same samples on different days and by different operators. Since the authors report that the assay is rapid and high throughput, they should be able to perform many/all of these assays or they should remove all references to ease of shipping, costs of the protein and the willingness to ship the material to any lab that wants it.

We have tried to clarify in the paper that HAT is not initially intended for clinical use, but for research purposes, including epidemiological studies. The referee is correct that internal controls with internationally validated standard sera would make harmonisation of results possible between experiments and between laboratories. The 20/130 WHO secondary standard serum is now available to the scientific community at large, and we have now recommended in the paper that CR3022 (that we provide) is titrated in each batch of assays, and this antibody can now be calibrated against serum 20/130 to provide universal standardisation.

Specifically concerns include:

1) There is no data on RBC donor-to-donor variations or RBC age impacts the assay sensitivity and specificity.

[Author] Response:

The 232 samples from figure 5 were tested a second time 34 days later with a different red cell donor. 99% of the measured titres were within one doubling dilution of each other. We have measured the control antibody CR3022 with multiple donors and not detected more than 1

doubling dilution difference in titre.

Action:

In response to the objection raised by this referee, we have selected three additional O-ve donors and performed quadruplicate titrations of a) monoclonal antibodies to each structure defined binding site on the RBD (FI-3A Class 1, Huang 2020; C121, Class 2, Robbiani 2020; S309 Class 3, Pinto et al 2020; and CR3022 Class 4, Huo 2020), and b) three positive human convalescent sera with different titres, as well as a negative serum.

Results:

Maximal variation of +/- 1 doubling dilution between all of the measurements can be seen on slides 14-16 in the pdf file provided. Those have already been discussed earlier, in our response to reviewer 1, comment 4.

Manuscript:

This additional data is now included in Supplementary Tables 3 and 4, and described in the text Lines 278-284

Reviewer 4 comment on response:

RBC age. The response does not include any assessment on RBC age as requested. That is, there is no description of whether fresh RBC's are required. The manuscript comments on how many wells can be assayed using a single isolation of RBCs from whole blood but lacks this critical piece of information.

We choose to refer to the appraisal of this point by referee 3 :

Comment 3:

I think the RBC age question is kind of stupid ...

We would add that as long as the CR3022 standard or 20/130 serum is in the assay any variation in sensitivity that might arise through the "age" of the red cells can be controlled for.

Donor to donor variation. The data presented partially addresses the concern. What is lacking is an indication of whether the quadruplicate experiments were presented independent replicates, done on the same day or the number of operators who performed the assay (not just the interpretation).

We feel that it was already pretty obvious that those were quadruplicate samples, i.e. four samples run in parallel by the same experimenter. To make things even clearer, we propose to specify : 'performed in quadruplicates (four samples run in parallel in the same experiment)' on lines 1013 (sup tab 3) and 1026 (sup tab 4).

2) Other than testing a set of pre-covid-19 serum samples, there is no formal evaluation on the impact of RBD targeting antibodies against endemic coronaviruses on the assay. Since the negative serum were not tittered against endemic coronaviruses, the specificity number may not be accurate.

[Author] Response:

This issue has already been discussed above, as an answer to the comment 3 from referee 1

[Author] Response [comment 3, referee 1]:

We tested 54 samples from patients with sepsis, all of which tested negative, and a total of 224 samples from other control donors giving a total of 278 negative samples with a single weak false positive result in the assays to assess sensitivity and specificity. Many of these donors are expected to have had high titre antibodies to influenza and other common infections, including common cold coronaviruses. Since those were clearly not detected in the test, we do feel that we have shown a reasonable data set to establish that, if there is some cross reactivity to other viruses, it is a very infrequent problem for HAT, and that the test is at least 99% specific. To do as

the referee asks would require a properly designed prospective study to collect blood samples from a large cohort of patients with respiratory illness for whom gold standard PCR diagnosis of each infection was available. This is simply not possible for this paper.

As a further elaboration on this issue, we know from published data that ~90% of adults are seropositive for OC43 (the most closely related of the common cold Betacoronaviruses), (Severance et al; <https://www.ncbi.nlm.nih.gov/pmc/articles/PMC2593164/>; and Anderson et al BioRxive 2020, <https://www.medrxiv.org/content/10.1101/2020.11.06.20227215v1>), and yet we are not detecting a high false positive rate in the pre-covid samples. This is because the great majority of this cross-reactivity is directed at the N protein (Severance et al) and the S2 domain of the Spike protein (Ng et al, <https://science.sciencemag.org/content/early/2020/11/05/science.abe1107>);. In the latter paper all of the cross-reactivity to the spike protein was absorbed out with the isolated S2 domain, but NOT by the S1 domain containing the RBD. This is very likely because there is very much less conservation of the RBD sequence between these viruses (~22% v 43% for the S2 domain). In our own work characterising monoclonal antibodies to SARS-CoV-2 (Huang et al, <https://www.biorxiv.org/content/10.1101/2020.08.28.267526v1>;) we found 5 out of 9 MAbs to S2 contained sequence features of memory cells (high level of somatic mutation) and cross-reacted on OC43 S2, and a similar proportion for N. For the RBD we found just 1 of 31 MAbs that cross-reacted with RBD from OC43. This single antibody was close to germline in sequence (i.e. Not derived from a memory B cell), and in any case failed to cross-link red cells in the HA test.

Finally, in our two assays to measure the operating characteristics of the HA test, the single false positive for each run was derived from a single well that showed a partial teardrop in one assay (scored negative) and loss of teardrop in the second (scored positive). In each case we investigated these by repeat measurements, and a titration with a reagent composed of IH4 without the RBD. In both cases the repeat assays showed only a partial teardrop in the first well of the titration, showing that whatever antibody was causing the reaction was low titre. In one case this weak reaction was clearly dependent on the presence of the RBD, i.e. could have been due to rare and very low titre cross-reactive antibodies between the RBD of a common cold virus and SARS-CoV-2. This is consistent with detection in 0.6-2% of precovid donors by Anderson et al. (see above) of binding activity to the RBD of SARS-CoV-2. In the second case the reaction was with the IH4 component. Lastly, we have heard from our colleague Prof Gathsaurie Malavige in Sri Lanka, who has been using the HA test successfully for some time, and found a single false positive in 110 pre-covid samples, from a patient with Dengue.

So, with this evidence in hand, we feel that false positive results in the HA test due to very low-level cross-reactivity between the RBDs of common cold betacoronaviruses and the RBD of SARS-CoV-2 will be rare (less than 1%), and can be distinguished by their low titre.

Reviewer 4 comment on response:

The betacoronaviruses HKU1 and OC43 are most closely related to SARS COV-2. However, exposure to the four endemic coronaviruses (i.e., 229E, HKU1, NL63 and OC43) in adults is high and the papers the authors refer to indicate that subjects have 60-92% seropositivity to the nucleoprotein for these species. While the data referenced by the authors shows a low level of cross-reactivity to SARS CoV-2 Spike in pre-pandemic serum, without testing the serum for activity against SARS CoV2 or endemic coronavirus, one should not assume the pre-pandemic serum is in and of itself a sufficient test. One particular concern would be an individual who has a high-titer to an endemic coronavirus spike protein following a recent infection and has developed high cross-reactivity to SARS CoV-2 Spike. This might cause a false positive. To test this, the authors should have tested pre-pandemic sera with high reactivity to the spike protein of each of the four human endemic coronaviruses. The

susceptibility of a diagnostic assay to conditions with the potential to impact its performance are important to define and understand prior to being broadly adapted. The requested analysis was not performed and therefore the authors did not respond this reviewer's concern.

See comment 4 by referee 3, starting with 'The reviewer appears to not know the current literature.'

a. Similarly, ideally more than one antigen would be used targeting epitopes that are specific to SARS-CoV-2. This would be expected to increase sensitivity and perhaps specificity.

[Author] Response:

This is a request that would be difficult to fulfil, and for the reasons outlined above would in all likelihood be pointless. The other possible "targeting epitopes" such as the S2 domain of Spike and the N protein are the most cross-reactive so would very likely reduce the assay specificity.

Reviewer 4 comment on response:

The authors response discusses the rationale for choosing RBD over other domains in Spike and using other virus protein in the response to reviewer #1, comment 3. The manuscript would be improved by adding this information and rationale to the manuscript. A paragraph on the rationale for choosing RBD was already present in the introduction, and we do not feel that we are in a position to discuss the advantage or disadvantage of this antigen compared to other ones from the SARS-2 virus since we have not studied them ourselves.

3) While I applaud that the assay controls for the presence of anti-RBC targeting antibodies, there is a striking lack of a negative and positive control serum. Well characterized reagents for this purpose should be included in the assay on every plate. This can be a home-made or commercially provide reagent. If end-users are going to be expected to provide the material for their assay, the ideal performance characteristics of the controls should be well described, to allow for end-users to perform the assay appropriately.

[Author] Response:

The reviewer is mistaken,— we described multiple negative control monoclonal antibodies to other areas of the spike (anti NTD and anti S2), and there are 199 negative sera in figure 4 and 79 negative controls in Figure 5. We described multiple positive control monoclonal antibodies to the RBD. We also include a positive control Monoclonal called CR3022 that is extremely well characterised with a crystal structure, and for which we have described the performance in terms of titration in detail in the paper, and which we provide free in the assay reagents we are distributing as a positive control. This antibody has been titrated with red cells from several donors and gives highly consistent results. We do not feel it is necessary to include the CR3022 "on every plate" but should be included with a titration with any batch of plates, to ensure that the IH4-RBD reagent is functioning correctly, which we stipulate in the methods.

Reviewer 4 comment on response:

While the monoclonal antibodies (mAbs) and serum used to develop the HAT are important, they are not used appropriately in the final assay as controls. First, mAbs are used in PBS and not in serum or plasma. This is not appropriate because the matrix can have an impact on the assay due to variations in serum/plasma proteins, fat content, hemolysis or other components present that are not in PBS. These physiochemical properties could impact the binding properties of the assay.

We have clearly stated that the assays using mAbs had been performed using diluted whole blood. The mAbs were thus not incubated in PBS, but in the same dilution of

plasma as for the intended use of HAT, similarly to all other experiments described in the paper

Next, the authors note the protein is stable following reconstitution for 1-2 weeks when refrigerated, there is no data presented on how that was determined and what the long-term stability is of the lyophilized material. Therefore, it is important to have a control that can accurately monitor for the loss of IH4-RBD integrity. In the single point and POCT assays it appears that the methods suggest using only one well to investigate the IH4-RBD integrity. It should be done as a dilution series in order to assess for a loss in antigen potency. The expected potency for the CR3022 in the assay should be provided as part of the acceptance criteria for the assay. Without this, the sensitivity of the assay might be impacted by IH4-RBD protein that has been degraded due to improper storage, handling or reconstitution by the operator.

We feel that the referee is, once again, being wilfully obtuse. How else than by titration curves could one determine that the activity of a reagent is the same after X number of days of storage at 4°C ? This must have been obvious to the other referees because none of them raised this issue.

Furthermore, while mAbs are easier to supply as a reagent they are not the ideal control for an assay. Ideally, a high and low positive control serum and negative control serum would be provided as a reagent and used as a dilution series in the assay. This will provide a positive control for the reagents and be used to confirm that the assay is performing as expected (i.e., within acceptance criteria). This is missing from that assay. Alternatively, mAb-spiked serum could be provided.

Additionally, ideally each plate would have a control associated with it to account for potential plate to plate variability and to confirm that the operator performed that assay as expected on each plate.

Finally, the titration method (page 25#2) does not mention a batch CR3022 control and, as currently written, also does not test for the presence of cold agglutinins by mixing serum with PBS and RBCs in the absence of RBD.

We, once again, underline that HAT is (for the time being) intended for research purposes. Any research team working on Covid will have easy access to positive and negative sera that can be used as controls, whereas providing it ourselves would be well nigh impossible since shipping of human blood materials is very restricted for H&S regulatory reasons. And, as we have already done earlier on, we point to the fact that the 20/130 WHO secondary standard serum is now available to the scientific community at large, and we have now recommended in the paper that CR3022 (that we provide) is titrated in each batch of assays, and this antibody can now be calibrated against serum 20/130 to provide universal standardisation.

4) Early sensitivity is being defined as the first five days of hospitalization, but the data should be discussed in time post-symptom onset as the time to presentation is variable and has the potential to bias the sensitivity higher or lower.

[Author] Response:

We feel that have made this issue very clear in the paper on line 517. It is not always possible to define the date of onset of symptoms, but the date of admission is an objective reality. The key point is that the test frequently becomes positive during the first five days of an average admission in a UK hospital, and is at least as sensitive as the Siemens Atellica test for detection of antibodies to the RBD in this context.

Reviewer 4 comment on response:

While the objective reality of an admission date is certainly easy to ascertain, the conclusions that are being drawn are potentially confounded by inaccurately using the date of admission. Presumably each of the subjects has a history and physical exam that was written at the time of admission to the hospital. Any good clinician would have asked their patients when they started to get ill and would have noted it in the associated NHS medical record. This should be easily obtained by a chart review and the data is likely to be available for the vast majority of subjects.

In the absence of reporting the data as requested, the authors should emphasize this a second time when discussion their conclusions (i.e., not just at the start of the section). See comment 7 of referee 3 for appraisal of this comment

5) Most critically, like other serologic assay's its highest sensitivity is in convalescent patients and is not surprisingly least sensitive during early illness. This limit's its use to population studies and for post-facto diagnosis of people who were not tested early on and is therefore, unlikely to have a significant impact on lowering disease transmission dynamics.

[Author] Response:

This statement is true of all serological testing, and does not warrant a detailed response.

Reviewer 4 comment on response:

The authors agree that the utility of this assay is in population studies and post-facto diagnosis and not as a test in the acute illness stage. Given this, the HAT in its current form has a low significance to the field.

The authors beg to differ on this point, and so do referees 2 & 3

Referee 2 (notes on report by referee 4)

It is clear R4 has taken a very careful look with the eyes of gold-standard assay validation. I just don't know if it is the right place to apply it in this instance.

With my reading this was not a clinical assay. Even if the assay is just used as an epidemiologic tool, it is still useful (also agreed to by R4). I do not agree (as stated by R4) that if the clinical utility of assay is potentially lower, the significance of this assay is low. This assay can help assess post infection assessment and vaccine development providing data that may otherwise never be collected (due to the low cost). In fact, when I read the paper, I was particularly pleased with (1) the low cost/simplicity and (2) free distribution to interested parties. The latter of which I imagined would lead to additional collaborations, assessments, and advancements of the assay. This is a free assay and in my opinion the benefit to using it is worth it.

Much of R4's issues appear to revolve around use in clinical setting. Ultimately, this seems to be more of philosophical debate: must assays that may have use in areas of limited resource be held to same standards of "1st world" settings.

- Perhaps the authors might lesson their suggestions of use or state due to the unique nature of the pandemic, validation will continue (basically a disclaimer). However, I feel I fall on the side of wider access to such assays, especially due to the severe limitations of any "perfect" assay and its likely much more limited distribution. While I am not directly in the clinical/diagnostic assay field, it is my understanding that a lot of diagnostic efforts to help low resource regions rely on techniques that may be less than gold standard assays.
- Perhaps the authors might note R4 concerns, such as RBC donor-to-donor variations, and then

address in follow up studies.

- Similar noting cross-reactivity mentioned by R4, and acknowledging to address in follow up studies.

Reasonable requests.

- R4's request for controls for dilution series experiments is reasonable if it will be the recommended setup. Similar, authors should indicate details of R4's comments on donor-to-donor variation.

Dilution experiments are not the intended setup for the HAT to be used. Those titration experiments were used to validate the performance of the assay, such as determining its sensitivity to increasing amounts of antibodies and such other variables.

- R4's request to discuss rationale for choosing RBD

A paragraph on the rationale for choosing RBD was already present in the introduction, and we do not feel that we are in a position to discuss the advantage or disadvantage of this antigen compared to other ones from the SARS-2 virus since we have not studied them ourselves.

- R4's following request are reasonable (suggested):

While the monoclonal antibodies (mAbs) and serum used to develop the HAT are important, they are not used appropriately in the final assay as controls. First, mAbs are used in PBS and not in serum or plasma. This is not appropriate because the matrix can have an impact on the assay due to variations in serum/plasma proteins, fat content, hemolysis or other components present that are not in PBS. These physiochemical properties could impact the binding properties of the assay.

Next, the authors note the protein is stable following reconstitution for 1-2 weeks when refrigerated, there is no data presented on how that was determined and what the long-term stability is of the lyophilized material. Therefore, it is important to have a control that can accurately monitor for the loss of IH4-RBD integrity. In the single point and POCT assays it appears that the methods suggest using only one well to investigate the IH4-RBD integrity. It should be done as a dilution series in order to assess for a loss in antigen potency. The expected potency for the CR3022 in the assay should be provided as part of the acceptance criteria for the assay. Without this, the sensitivity of the assay might be impacted by IH4-RBD protein that has been degraded due to improper storage, handling or reconstitution by the operator.

Furthermore, while mAbs are easier to supply as a reagent they are not the ideal control for an assay. Ideally, a high and low positive control serum and negative control serum would be provided as a reagent and used as a dilution series in the assay. This will provide a positive control for the reagents and be used to confirm that the assay is performing as expected (i.e., within acceptance criteria). This is missing from that assay. Alternatively, mAb-spiked serum could be provided.

Additionally, ideally each plate would have a control associated with it to account for potential plate to plate variability and to confirm that the operator performed that assay as expected on each plate.

Finally, the titration method (page 25#2) does not mention a batch CR3022 control and, as currently written, also does not test for the presence of cold agglutinins by mixing serum with PBS and RBCs in the absence of RBD.

The replies to those various points have already been provided above

Referee 3 (comments on report by referee 4)

Comment 1:

I think a compromise could be reached to state that the reagent cost is 0.27 pence, and that clinical use could be \$2-5, including blood draw, staff labor, and disposable plates.

Agreed, and done (see line 483)

-the next comment here kind of misses the point; this HAT test doesn't need to be better than the commercial tests, just cheaper

Indeed 😊

-the comments about sensitivity/specificity of serology tests in general are outside the scope of this manuscript. All tests suffer from limitations of positive predictive value, so I don't really get the relevance of arguing/debating that in this manuscript, which focuses on a novel technology, with great low cost.

Indeed 😊

-The reviewer arguing about the clinical utility of serology tests is again missing the point of the paper, and outside the scope of the manuscript. The reality is that antibody testing has proven useful for judging who may be protected from infection, who may benefit from convalescent plasma or recombinant monoclonal antibody therapy, who may not pose an infectious risk during COVID-19 infection and help prioritize vaccination resources. It's not simply epidemiology. Lots of people are getting ELISA tests for antibodies at this moment, and those are costing a significant amount of money, so it's not just for epidemiology.

Again, in my opinion, saying that antibody tests are stupid so this paper is stupid is more of a personal opinion, and not relevant to the data findings in the work.

Indeed 😊

Comment 2:

I thought the controls were appropriate and do not have issue with them. They did rigorous sensitivity and specificity testing with clinical samples, and in the earlier work used PBS controls. Clearly the assay works as intended and the authors don't need to repeat the same controls in every figure.

I don't think this HAT assay is meant to be shippable tomorrow as a clinical test. The paper focuses on characterizing the technology and presents data suggesting clinical utility. Subsequent testing will enable this to occur. Remember, this is an academic lab publishing this, not a diagnostic testing company. The Nature communications paper isn't an FDA submission packet. I think it's outside the scope of the authors expertise.

We could not have put it better ourselves. Thank you

Comment 3:

I think the RBC age question is kind of stupid. The actual clinical assay would be using dilute whole blood from donors anyways. The authors used serum and RBC's from various donors mostly because only COVID-19 serum is banked right now. Any blood bank reagent red cells used furthermore are already a sold product, so this isn't something that needs to be reinvented.

I'm failing to see the concern of the reviewer on donor to donor variation. The fusion protein targets a conserved epitope in the human population. Not sure why the reviewer is worried? Seems like asking questions to ask questions.

We could not agree more ...

Comment 4:

The reviewer appears to not know the current literature. There is a lot of data investigating cross-reactivity of RBD against the other endemic viruses (229E, HKU1, NL63 and OC43). Stating some hypotheticals by high titer cross-reactivity has not been born out in any of the other

serology papers I've seen, including the Nature Medicine one. So again, this suggestion seems to be pure speculation with no literature to back it up, and I'm not even sure how the authors would obtain it. Moreover, this risk is for all serology tests, so doesn't need to be litigated by this one group.

<https://www.nature.com/articles/s41591-020-0913-5>

No further comment needed here.

Comment 5:

I think a compromise would be add the rationale for choosing RBD. Would just require 1-2 sentences and a couple of references to the paper.

A paragraph on the rationale for choosing RBD was already present in the introduction, and we do not feel that we are in a position to discuss the advantage or disadvantage of this antigen compared to other ones from the SARS-2 virus since we have not studied them ourselves.

Comment 6:

I think this debate is sort of about the potential commercial rollout of the test, and what the best positive control for such a test to be. Arguing about this is kind of futile, since it will be cost considerations for any company that picks this up for formal regulatory approval. It might be lost on reviewer #4 that academic labs would benefit from a cheap test to do titration assays for COVID-19 samples in a research setting. In any case, a question of cost/money and exact controls for the commercial test shouldn't hold up publication.

I think the authors could add one more sentence to their protocol about having a control without RBD fusion protein, and just PBS and patient RBC/serum.

The importance of running a negative control with every sample tested is already stressed numerous times in the paper, including in figures 4 and 6 which show examples of such tests.

Comment 7:

I would disagree that it's easy to always know from patient data when the exact symptoms start. I am a clinician and have discussed with many colleagues, including friends and family, that it was vague when they might have started feeling bad. Even if one did do chart review, it would take a long time to acquire all of this information, and it may still be inaccurate. I sympathize more with the authors here, that using known admission dates is an objective criteria. Moreover, the time of seroconversion has already been studied in other manuscripts, and is outside the focus of the current technical manuscript. I think the authors noting this limitation once was enough.

Could not agree more, thanks 😊

Comment 7:

I'm not sure why this reviewer has a very narrow mind when it comes to the utility of serology tests. The lack of widely available serology tests has been a problem during the pandemic, preventing their application for different stages and/or aspects of treatment. Stating an entire testing category has low significance is an opinion to me. If such low significance, why did Nature Medicine publish an entire paper on an ELISA assay.

No comment needed here, apart from thanking referee 2 and 3 once more for their support, hard work, and dedication.